# Kernel Perspective Review Response 1, 2, & 3

January 1, 2024

## Summary

We would like to thank the reviewers for their efforts towards the betterment of our manuscript. The reviewers have looked for typographical and theoretical improvements to our manuscript, and we sincerely appreciate it. Below are a summary of some of the remarks made by the reviewers, and our efforts to adjust the manuscript to address them.

## 1 Response to Reviewer sZBJ

**Reviewer sZBJ:**
The text is hard to read. The motivation of this study and its practical implications are missed.

The manuscript structure requires significant revision since the beginning of the text is just a review of advanced concepts from functional analysis and ergodic theory placed there without any explicitly stated motivation. The further text is also hard to read and extract the main contribution of the authors. I suggest highlighting the main intuition behind the abstract math concepts and using them to help the reader catch the addressed issues, the proposed solution, and the practical gain if any.

**Author's Response:** We thank the reviewer for pointing out these oversights in our manuscript and intend on correcting them by adding in an introduction. The introduction, which can be seen in the included revised manuscript, mentions the following:

- The main contributions of the author's: Establish a new kernel perspective for Koopman analysis and DMD, addressing the properties that are assumed to hold for Koopman operators, and providing a DMD algorithm for densely defined Koopman operators.

- The need for an alternative perspective to that of ergodic theory. This will aid us in motivating our introduction to the functional analysis and ergodic theory concepts mentioned by the reviewer.

**Reviewer sZBJ:**
The single numerical test is presented
This test does not show any (visual) difference between the proposed kernel and RBF

**Author's Response:**
A single numerical test was presented as a way of showing that the results obtained through our new algorithm are similar to those obtained by previous DMD algorithms. However, in our revision we have added another numerical example. If the reviewers feel strongly about inclusion of these results in the manuscript, we are open to the idea, but we reiterate that we do not claim to have improved upon the KDMD algorithm in any practical sense. The new derivation provides a much better understanding of previously overlooked theoretical limitations of KDMD, but computationally, the algorithms are nearly identical. That being the case, we believe that a proof-of-concept simulation that shows performance at-par with KDMD is sufficient.

For our second numerical experiment we chose to use data from [Jakob et al.] ([2021]). Specifically we chose a data set that corresponds to 2-dimensional turbulent flow with a Reynolds number of 4092.2162409351754 and a Kinematic viscosity of $5.4501926528485274\exp-05$. The data as well as the experiment conducted on the data is further described in the revised paper.

With regards to the second comment, we did not propose a new kernel function, but rather a new DMD algorithm that was based on a novel kernel perspective of Koopman analysis. We believe that this confusion might have been due to the lack of a clear introduction as mentioned by the reviewer.

**Reviewer sZBJ:**
Experiments do not report any quantities to formally compare the considered DMD-like approaches

The single numerical test of the proposed new DMD framework is not enough for a solid justification of its practical importance. In addition, the presented test is not equipped with any quantities to estimate the approximation accuracy of the prediction generated by the baseline and the proposed kernel DMD. I do not observe any visual difference between generated snapshots. Therefore, the practical importance of the introduced kernel is not clear.

**Author's Response:**
In order to address the reviewer's concerns we will add a more formal metric for comparing the snapshots that were obtained from reconstruction to those from the original data set. We will use the 2-norm of the error between the true snapshots and the snapshots generated by our DMD method. Additionally, we compare the 2-norm of the error obtained from our algorithm to that obtained when using the kernel DMD algorithm proposed in Williams et al. (2015b).

**Reviewer sZBJ:**
No ablation study in the experiments: how the proposed method is robust to the hyperparameter $\mu$?

**Author's Response:**
It was not the goal of the author's to solve hyperparameter tuning problems as these will arise both in the method that we have created and in previous methods. When it comes to the variability in results when hyperparameters are changed there is no reason to think that our method does any better or worse than previous DMD methods that rely on the user defining a kernel.

**Reviewer sZBJ:**
The computational complexity (both asymptotic and runtime) of the introduced modification of Kernel DMD is completely ignored

**Author's Response:**
The computational complexity for this method can be shown to be at most $O\left((m-1)^3\right)$ where $m$ represents the number of snapshots. The reason for this computational complexity is that there are two inversions of an $m-1\times m-1$ matrix and there is an eigendecomposition which must be computed for a $m-1\times m-1$ matrix.

**Reviewer sZBJ:**
Moreover, although the authors provide artificial counterexamples for the Koopman operator properties, they do not illustrate what practical implications can be observed due to the violation of these properties. Thus, the practical motivation of the manuscript is unclear, please add the explicit problem statement section, where the quality metric is introduced and then used in the experiment section to support the gain from the proposed approach.

**Author's Response:**
Although we agree that a study that explores the practical implications of applying DMD type methods when violating the properties of the Koopman operators in RKHSs would be interesting, it was not the purpose of our paper. In presenting this new perspective we aimed to set a rigorous theoretical framework for Koopman based DMD over kernel spaces. Such a framework required that we highlight the theoretical limitations of Koopman based DMD.

**Reviewer sZBJ:**
Check the typesetting: many typos, and wrong formatting. Unfortunately, without the line numbers listing typographic issues is impossible.

**Author's Response:**
We thank the reviewer for pointing this out to us and we will address this in our revision.

**Reviewer sZBJ:**
If the authors provide the modification of the DMD algorithm, it would be helpful to provide the pseudocode on how to transform snapshots to generate dynamic modes and how to use them further in forecasting

**Author's Response:**
At the request of the reviewer we have provided a pseudocode for the DMD algorithm that results from the novel perspective that we have presented.

---

**Algorithm 1:** Pseudocode for the novel kernel perspective based DMD algorithm. Upon obtaining the Koopman modes, the approximate eigenfunctions, and the eigenvalues, equation (8) is used to compute $x_{i+1}$.

---

**Input:** Snapshots, $X = \{x_1, x_2, ..., x_m\}$
**Input:** Kernel function $K : \mathbb{R}^n \times \mathbb{R}^n \to \mathbb{R}$ of an RKHS
Step 1 Compute the Gram matrix, $G = (K(x_i, x_\ell))_{i,\ell=1}^{m-1}$
Step 1.5 If $G$ is ill-conditioned, then set $G = G + \epsilon I_{m-1}$
Step 2 Compute the interaction matrix, $\mathcal{B} = (K(x_i, x_\ell))_{i=2,\ell=1}^{i=m,\ell=m-1}$
Step 3 Compute the finite rank representation of the Koopman operator, $[P_\alpha \mathcal{K}_F]_\alpha^\alpha = G^{-1}\mathcal{B}$
Step 4 Compute eigenvalues, $\lambda_j$, and eigenvectors, $v_j$, of $[P_\alpha \mathcal{K}_F]_\alpha^\alpha$
Step 5 Compute the matrix of Koopman modes, $\hat{\xi} = X (V^T G)^{-1}$
Step 6 Compute approximate eigenfunctions, equation (7)
**Output:** Koopman modes, $\hat{\xi}_j$ for $j = 1, \ldots, m-1$
**Output:** Approximate eigenfunctions, $\hat{\varphi}_j(x)$ for $j = 1, \ldots, m-1$
**Output:** Eigenvalues $\lambda_j$ for $j = 1, \ldots, m-1$

---

**Reviewer sZBJ:**
The link to the author's implementation of the proposed approach will be also very helpful for the community and simplify extensive testing of the proposed DMD approach in other, more complicated use cases

**Author's Response:**
The final version of the paper, if accepted, will include a link to the code. The authors are unable to add links to the code in the attached revision due to the double blind nature of the review process.

# 2 Response to Reviewer RXz3

**Reviewer RXz3:**
It is difficult to assess where in the paper the contribution starts. The manuscript presents background material until (if I am not mistaken) page 7, and then the novel part starts in Sec 4. I am unsure if I captured the contribution to the extent that I cannot state that the contribution is at the bar of TMLR

**Author's Response:**
The novel parts of the paper start in section 3.2.3 where we proved that the Koopman operator over the Gaussian RBF's native space is only bounded when it corresponds to discrete dynamics that are affine. However, we are not aware of a critical analysis of various proposed properties of Koopman operators that has been assembled in such a way as in the previous section.

We thank the reviewer for informing us of their confusion and we aim to correct the problem by adding in an introduction that will highlight the contributions of the paper as was stated with the response to a comment made by the other reviewer.

**Reviewer RXz3:**
Sec 4 is said to "propose a method". However, it is not clear what this method is, how it differs from others, and its motivation. This is supposed to be the core contribution of the article; however, it is somewhat drowned out by the rest of the paper.

**Author's Response:**
As a result of having introduced a new kernel perspective of Koopman operator based analysis, a new theoretical framework for kernel-based dynamic mode decomposition (DMD) was presented in section 4, which leads to a simplified algorithm as compared with previous kernel-based DMD algorithms. We believe that the addition of pseudocode may make the method more clear to the reader, but the title of section 4 indicates that the method is a DMD method with Koopman operators over reproducing kernel Hilbert spaces.

The reviewer mentions that the core contribution is drowned out by the rest of the paper, a statement with which we the author's would disagree. The prior sections motivate the examination of densely defined Koopman operators over RKHSs, which is a core element of our theoretical approach. To arrive at our method requires the use of the properties of Koopman operators and kernel functions over reproducing kernel Hilbert spaces. Additional motivation for this method is mentioned in the first paragraph of section 4 where we mention that we are providing a simplification of prior kernel-based dynamic mode decomposition methods.

The simplification mentioned in the first paragraph of section 4 also answers the reviewers question regarding the difference between our method and other methods. Additionally, the majority of section 2 was used to show that prior methods for DMD which relied on ergodic theory resulted in DMD based models that were only accurate almost everywhere. This is in contrast to the DMD based model that results when convergence is achieved by our method which is accurate point-wise everywhere.

**Reviewer RXz3:**
The experimental validation is far from convincing. The authors only show a single synthetic example and then claim that purely from visual inspection, this example validates their proposal. This is a jumping conclusion not supported by the appropriate evidence.

**Author's Response:**
Since our goal was to describe a different perspective of Koopman based DMD methods which fully took advantage of the RKHSs in which the kernel functions reside, we found it sufficient to provide a single benchmark example. However, we have added another experiment to bolster our claim as well as provide a more rigorous quantitative assessment to show that our method performs comparably to the prior DMD methods. More information is provided in the Numerical Experiments section of the revised paper.

**Reviewer RXz3:**
Not only does the paper lack a quantitative assessment of the method, but the experiment does not consider benchmarks or other methods for comparison.

**Author's Response:**
As mentioned above, we will address this concern by adding a more formal metric for comparing the snapshots that were obtained from reconstruction to those from the original data set. We will use the 2-norm of the error between the true snapshots and the snapshots generated by our DMD method. Additionally, we compare the 2-norm of the error obtained from our algorithm to that obtained when using the kernel DMD algorithm proposed in Williams et al. (2015b).

**Reviewer RXz3:**
A particularly serious issue with this paper is that the Conclusion is almost a verbatim copy of the Abstract. Please avoid this practice

**Author's Response:**
We thank the reviewer for pointing out this oversight and have included a new abstract below.

**Abstract:**
The purpose of the new DMD algorithm developed in this paper is to show that DMD methods very similar to KDMD emerge naturally out of a finite rank representation of the Koopman operator. It should be noted that the developed algorithm, while derived in a different way than traditional KDMD, involves computations that are nearly identical to KDMD, and as such, is not expected to offer any performance benefits over KDMD. Moreover, the algorithmic development of the present method does not invoke feature space representations and infinite matrices as in Williams et al., rather this method uses directly the properties of Koopman (or composition) operators and kernel functions. By doing so, this makes the theoretical dependencies of kernel based DMD methods transparent as densely defined operators over infinite dimensional kernel spaces. In order to present this new kernel perspective of Koopman analysis, the manuscript first introduces reproducing kernel Hilbert spaces (RKHSs) and examines the properties of Koopman operators over said spaces. Additionally, the examination of these properties led to the proof that the Koopman operator over the Gaussian RBF's native space is only bounded when it corresponds to discrete dynamics that are affine.

# 3 Response to Reviewer jovM

**Reviewer jovM:**
What is the point of the discussion about universal kernels? It is never mentioned past the first section, and there is no consistency theorem, nor even a discussion that needs to have this notion introduced.

**Author's Response:**
Universal kernels were defined in Steinwart and Christmann (2008) in the following way:

**Definition 1.** A continuous kernel $k$ on a compact metric space $X$ is called **universal** if the RKHS $H$ of $k$ is dense in $C(X)$, i.e., for every function $g \in C(X)$ and all $\epsilon > 0$ there exists an $f \in H$ such that

$$\|f - g\|_\infty \leq \epsilon$$

The universality property arises in the search for Koopman "eigenfunctions," which, given a particular Koopman operator and kernel space, might not exist, since the existence of an eigenfunction depends on the Hilbert space. However, the formal equation, $\varphi(F(x)) - \lambda\varphi(x) = 0$ may still hold for some continuous function $\varphi$. If we are working over a RKHS corresponding to a universal kernel, then for any given compact set and $\varepsilon > 0$, there is a function $\tilde{\varphi} \in H$ such that $|\varphi(x) - \tilde{\varphi}(x)| < \varepsilon$ for all $x$ in that compact set, which provides for an "approximate" eigenfunction.

**Reviewer jovM:**
Could you explicit the difference between [Williams 2015a] and your approach for DMD?

Clarify the relation with prior work

**Author's Response:**
The biggest difference between the work of Williams et al and our own work is that Williams is largely heuristic manipulations of infinite dimensional matrices. In Williams, any properties or relations between the kernel spaces and the dynamics are ignored, where there is no discussion of the boundedness or densely definedness required of operators according to their methods. This manuscript endeavors to make transparent the relationship between Koopman operators, DMD, and kernel spaces, where we demonstrate explicitly the sorts of dynamics for which we can expect Koopman operators to be bounded or compact, and even in the case where the Koopman operator has neither property, we demonstrate that very similar computations can still be theoretically justified by examining relationships between the densely defined Koopman operators and kernel functions.

**Reviewer jovM:**
What is the argument for saying $\xi_{j,M}$ exist, without assumptions on the family of $(\phi_j)_{j=1}^{\infty}$?

**Author's Response:**
For any finite collection of linearly independent vectors, we can compute the projection of another vector onto their span in an inner product space (this can be done in part by inverting the gram matrix for the finite collection of vectors). For an infinite collection of eigenfunctions, there may not be a fixed collection of modes that correspond to each individual eigenfunction, unless those eigenfunctions are orthogonal. However, if one projects onto a finite sub-collection of linearly independent eigenfunctions, $\{\phi_1, \ldots, \phi_M\}$, then for that finite collection, the modes are uniquely determined. That's what the $\xi_{j,M}$ represent.

**Reviewer jovM:**
The full state observable is denoted $g_{id}$ but it does not depend on $d$, nor on $i$.

The vector-valued eigenfunction is denoted $\phi_{s,i}$ but it does not depend on $i$ either.

**Author's Response:**
The full state observable is also known as the identity function and thus we chose to represent it as $g_{id}$. In this case the subscript does not indicate a dependence on $i$ or on $d$, but rather serves as a short hand for the word identity. However, in order to make it more clear we have decided to change the text style for the subscripts used for the full state observable. The function will now appear as follows: $g_{\mathrm{id}}$

It is mentioned in the manuscript that the vector valued eigenfunctions are defined as $\phi_{s,i} := \tilde{\phi}_s e_i$. Therefore, $\phi_{s,i}$ is dependent on $i$.

**Reviewer jovM:**
The critical passage that explicits $[P_\alpha \mathcal{K} F]_\alpha^\alpha$ deserves more explanation, a lemma would be welcome.

Improve the presentation of the derivations.

Improve the part about vector-valued RKHSs.

**Author's Response:**
In this response we are including all the details that were left out of the paper. Some of these are relatively elementary concepts in linear algebra and the matrix representation of a linear transformation over a finite dimensional vector space, while others invoke the general theory of operator valued kernel spaces. We give a complete description of how we obtained the matrix representation of the finite rank representation of the Koopman operator, and then we describe why general operator valued kernels make DMD algorithms

infeasible because of poor scaling of the inversion problem. This scaling issue is ameliorated through the selection of diagonal kernel operators, which can either eliminate the scaling of the inverse problem with respect to the state dimension or change it to scale linearly.

The authors would like to apologize for the tutorial like response below, but we include it for maximum clarity.

Recall that in a real valued Hilbert space, the projection of a function $g$ onto a collection of linearly independent basis functions, $u_1, \ldots, u_M$, is a linear combination of those functions, $Pg = \sum_{j=1}^{M} w_j u_j$ where the weights $w_j$ may be determined via

$$
\begin{pmatrix} w_1 \\ \vdots \\ w_M \end{pmatrix} = \begin{pmatrix} \langle u_1, u_1 \rangle_H & \cdots & \langle u_1, u_M \rangle_H \\ \vdots & \ddots & \vdots \\ \langle u_M, u_1 \rangle_H & \cdots & \langle u_M, u_M \rangle_H \end{pmatrix}^{-1} \begin{pmatrix} \langle g, u_1 \rangle_H \\ \vdots \\ \langle g, u_M \rangle_H \end{pmatrix}.
$$

Let $E$ be a linear transformation between two finite dimensional vector spaces $V$ and $W$. Suppose that $\alpha = \langle \alpha_1, \ldots, \alpha_N \rangle$ is an ordered basis for $V$, and suppose that $\beta = \langle \beta_1, \ldots, \beta_M \rangle$ is an ordered basis for $W$. The matrix representation of the linear transformation, $E$, with respect to the ordered bases $\alpha$ and $\beta$ is denoted as $[E]_\alpha^\beta$, where the $j$-th column of this matrix is the weights of the vector $E_{\alpha_j}$ corresponding to the ordered basis $\beta$.

Now, let's address the current context of learning a dynamical system, $F : \mathbb{R}^n \to \mathbb{R}^n$, from a sequence of snapshots, $\{x_0, \ldots, x_M\} \subset \mathbb{R}^n$. Let $H$ be a RKHS with kernel $k_y(x) = k(x, y)$, and take $\alpha = \langle k_{x_0}, \ldots, k_{x_{M-1}} \rangle$ to be our ordered basis. Define $P_\alpha$ to be the projection onto the span of $\alpha$. The finite rank representation of the Koopman operator, $\mathcal{K}_F : D(\mathcal{K}_F) \to H$ with $D(\mathcal{K}_F) := \{g \in H : g \circ F \in H\}$. We will assume that $\mathcal{K}_F$ is densely defined and that the kernel functions are in the domain of $\mathcal{K}_F$.

We are taking the finite rank representation of $\mathcal{K}_F$ to be the operator given as $\tilde{\mathcal{K}}_F := P_\alpha \mathcal{K}_F P_\alpha$. Note that this operator is finite rank, since $\alpha$ is finite. If we restrict our domain to $span(\alpha)$ then we can make a matrix representation of $\tilde{\mathcal{K}}_F$ with respect to the $\alpha$ basis in both the domain and the range of the operator. The $j$-th column of this matrix consists of weights corresponding to the $\alpha$ basis obtained from $\tilde{\mathcal{K}}_F k_{x_j}$. These may be obtained by projecting $\tilde{K}_F k_{xj}$ onto $\alpha$.

In other words, each column of the matrix $[\tilde{\mathcal{K}}_F]_\alpha^\alpha$ may be determined via

$$
\begin{pmatrix} \langle k_{x_1}, k_{x_1} \rangle_H & \cdots & \langle k_{x_1}, k_{x_{M-1}} \rangle_H \\ \vdots & \ddots & \vdots \\ \langle k_{x_{M-1}}, k_{x_1} \rangle_H & \cdots & \langle k_{x_{M-1}}, k_{x_{M-1}} \rangle_H \end{pmatrix}^{-1} \begin{pmatrix} \langle \tilde{\mathcal{K}}_F k_{x_j}, k_{x_1} \rangle_H \\ \vdots \\ \langle \tilde{\mathcal{K}}_F k_{x_j}, k_{x_{M-1}} \rangle_H \end{pmatrix}
$$

Note that $\langle \tilde{\mathcal{K}}_F k_{x_j}, k_{x_i} \rangle_H = \langle P_\alpha \mathcal{K}_F P_\alpha k_{x_j}, k_{x_i} \rangle_H = \langle \mathcal{K}_F k_{x_j}, k_{x_i} \rangle_H = k_{x_j}(x_{i+1})$ since $P_\alpha$ is selfadjoint and also acts as the identity on $span(\alpha)$. Note also that $\langle k_{x_j}, k_{x_i} \rangle_H = k(x_i, x_j)$.

Now since this same inverse of the Gram matrix is applied to each and every column of $[\tilde{\mathcal{K}}_F]_\alpha^\alpha$, it can be pulled out to the left and we have the final representation given in the manuscript.

Now for the scalar valued case, we decompose, individually, the coordinate functions with regard to the eigenfunctions obtained from the finite rank approximation of the Koopman operator and ultimately receive a decomposition of the full state observable by creating an ensemble of the individual decompositions.

If we want to decompose the full state observable all at once, then we need to appeal to vector valued RKHSs, which produce an operator valued kernel function. Given a vector valued RKHS, $H$, consisting of functions that map a set $X$ to a Hilbert space $\mathcal{Y}$, there is for each $x \in X$ and $\nu \in \mathcal{Y}$ a function $K_{x,\nu} \in H$ such that $\langle g, K_{x,\nu} \rangle_H = \langle g(x), \nu \rangle_\mathcal{Y}$. The mapping $\nu \to K_{x,\nu}$ is linear, and hence, we express the kernels as $K_x \nu$ where $K_x : \mathcal{Y} \to H$ is a bounded operator. In this case, for each $g \in H$, $K_x^* g = g(x)$. The operator valued kernel associated with $H$ is given as $K(x,y) := K_x^* K_y : \mathcal{Y} \times \mathcal{Y} \to H$, and note that when $\mathcal{Y} = \mathbb{R}^n$, this operator valued kernel is a matrix whose entries are scalar valued functions.

When a vector valued RKHS, $H$, is obtained from a scalar valued RKHS, $\tilde{H}$ with kernel $k(x,y)$, as $H = \tilde{H}^n$ where the inner product of two elements, $g = (g_1, \ldots, g_n)^T$ and $h = (h_1, \ldots, h_n)^T$, from $H$ is given as $\langle g, h \rangle_H = \sum_{j=1}^n \langle g_j, h_j \rangle_{\tilde{H}}$, then the operator valued kernel is given as $k(x,y)I_n$, which has many convenient properties for computation.

Here the Koopman operator is defined just as it would be for a scalar valued space: $\mathcal{K}_F g = g \circ F$, and the difference here is that $g$ is vector valued. Similar relationships between the Koopman operator and the kernel functions still hold, where $\langle \mathcal{K}_F g, K_x \nu \rangle_H = \langle g(F(x)), \nu \rangle_{\mathcal{Y}} = \langle g, K_{F(x)} \nu \rangle_H$, hence $\mathcal{K}_F^* K_x = K_{F(x)}$.

Now given a collection of snapshots, $\{x_1, \ldots, x_M\}$, we have the associated operator valued kernels $K_{x_1}, \ldots K_{x_M}$. For decomposition of the full state observable, $g_{id}(x) = x$, we need to interface with the Hilbert space and construct a finite rank approximation of $\mathcal{K}_F$ as we did above. This means we need to construct a subspace within $H$, say spanned by a collection of basis functions $\alpha$, so we can write $\tilde{\mathcal{K}}_F = P_\alpha \mathcal{K}_F P_\alpha$.

The selection of the basis $\alpha$ is less constrained than in the scalar valued case, where we only had the kernels centered at the snapshots to consider. In this case, we have the snapshots, but also complete freedom of the choice of $\nu \in \mathbb{R}^n$ to select each $K_{x,\nu} = K_x \nu$. A natural choice is to select $\nu$ from the standard basis of $\mathbb{R}^n$. This leads to $\alpha = \langle K_{x_0, e_1}, \ldots, K_{x_{M-1}, e_1}, K_{x_0, e_2}, \ldots, K_{x_{M-1}, e_2}, \ldots, K_{x_0, e_n}, \ldots, K_{x_{M-1}, e_n} \rangle$, which is a huge basis for problems such as the cylinder flow example, where $n$ is of the order $80,000$. Consequently, the gram matrix corresponding to this basis is a square matrix of dimension $M \cdot n$.

So for arbitrary operator valued kernels, the DMD algorithm presented in this paper would have to invert an $M \cdot n$ dimensional matrix, and the computation time would scale with the dimension of the space as $O((M \cdot n)^3)$ using standard inversion algorithms. This scaling would make benchmarks such as the cylinder flow example completely infeasible.

Computational feasibility can be achieved through the judicious selection of the kernel operator. Here we leveraged a kernel operator of the form $K(x,y) = k(x,y)I_n$ with $k$ a scalar valued kernel function. In this case, $\langle K_x \nu, K_y \omega \rangle_H = k(x,y) \langle \nu, \omega \rangle_{\mathbb{R}^n}$ and the two functions $K_x \nu$ and $K_y \omega$ are orthogonal when $\nu$ and $\omega$ are orthogonal in $\mathbb{R}^n$. This means that the Gram matrix composed of the $\alpha$ given above is going to be a block diagonal matrix with each block being the Gram matrix corresponding to the scalar valued kernel. Thus only one matrix of size $M$ must be inverted, and each dimension treated individually. Significantly, the inversion problem no longer scales with the dimension.

Alternatively, if we instead use a different scalar valued space for each dimension, we would get $n$ different Gram matrices to invert. The complexity would scale with the dimension, but linearly rather than cubicly.

So while it is possible to treat this with general operator valued kernels, this would only be expected to work for small state dimensions, and handling fluid flow problems (which is a typical application of DMD methods) would not be possible.

# References

J. Jakob, M. Gross, and T. Günther. A fluid flow data set for machine learning. `https://doi.org/10.3929/ethz-b-000515488`, 2021. Accessed: 2023-12-29.

I. Steinwart and A. Christmann. *Support vector machines*. Springer Science & Business Media, 2008.

# The Kernel Perspective on Dynamic Mode Decomposition

**Anonymous authors**

## Abstract

The purpose of the new DMD algorithm developed in this paper is to show that DMD methods very similar to KDMD emerge naturally out of a finite rank representation of the Koopman operator. It should be noted that the developed algorithm, while derived in a different way than traditional KDMD, involves computations that are nearly identical to KDMD, and as such, is not expected to offer any performance benefits over KDMD. Moreover, the algorithmic development of the present method does not invoke feature space representations and infinite matrices as in Williams et al., rather this method uses directly the properties of Koopman (or composition) operators and kernel functions. By doing so, this makes the theoretical dependencies of kernel based DMD methods transparent as densely defined operators over infinite dimensional kernel spaces. In order to present this new kernel perspective of Koopman analysis, the manuscript first introduces reproducing kernel Hilbert spaces (RKHSs) and examines the properties of Koopman operators over said spaces. Additionally, the examination of these properties led to the proof that the Koopman operator over the Gaussian RBF's native space is only bounded when it corresponds to discrete dynamics that are affine.

## 1 Introduction

Dynamic mode decomposition (DMD) has been gaining traction as a model-free method of making short-run predictions for nonlinear dynamical systems using data obtained as snapshots of trajectories. DMD and its variant, extended DMD, have proven effective at extracting underlying governing principles for dynamical systems from data, and reconstruction methods using DMD allow for the modeling of nonlinear dynamical systems as a sum of exponential functions, which is analogous to models obtained for linear systems (Kutz et al., 2016b).

DMD is closely connected to the Koopman operator corresponding to discrete time dynamical systems (Schmid, 2010). The Koopman Operator $\mathcal{K}_F$ over a Hilbert function space, $H$, is a composition operator corresponding to the discrete time dynamics, $F : \mathbb{R}^n \to \mathbb{R}^n$, acting on functions in the Hilbert space. Specifically, $\mathcal{K}_F g = g \circ F$. In the context of discrete time dynamical systems, $F$ is the discrete time dynamics that maps the current state of the system to a future state $x_{i+1} = F(x_i)$. Frequently, the function $g \in H$ is referred to as an *observable*.

DMD aims to obtain a finite rank representation of the Koopman operator by studying its action on the full state observable (i.e. the identity function) (Schmid, 2010). Koopman operators over reproducing kernel Hilbert spaces (RKHSs) were studied to take advantage of infinite dimensional feature spaces to extract more information from the snapshots of a system in Williams et al. (2015a). This perspective also enacts a dimensionality reduction by formulating the DMD method in a reproducing kernel Hilbert space (RKHS) framework and implicitly using the kernel trick to compute inner products in the high-dimensional space of observables. In Williams et al. (2015b), it is shown that kernel-based DMD produces a collection of Koopman modes that agrees with other DMD results in the literature.

The introduction of kernel based techniques for Koopman analysis and DMD yields a new direction for Koopmanism. Traditionally, the theoretical foundations of Koopman operators, and their implementation in the study of dynamical systems, have been rooted in ergodic theory. When working with functions that are

$L^2$ with respect to an ergodic measure, the Birkhoff ergodic theorem guarantees that time averages under the operation of the Koopman operator will converge almost everywhere, or with probability 1 Budišić et al. (2012). This convergence gives invariants for the Koopman operator in $L^2$ with respect to the invariant measure corresponding to the dynamics, and the Wiener Wintner theorem can be leveraged to realize the spectrum of the operators.

This is somewhat unsatisfying, since for a deterministic continuous time dynamical system, the ultimate result of discretization and Koopman analysis is probabilistic. Moreover, since the point-wise convergence given by the Birkhoff theorem holds almost everywhere, as opposed to everywhere, there is no guarantee that convergence will be realized at a given point. Furthermore, since all numerical methods deal with finitely many data points, convergence is impossible to verify.

The new perspective added by kernel methods is that of approximations. Universal RKHSs, such as those corresponding to Gaussian RBFs and Exponential Dot Product kernel functions, have the ability to approximate any continuous function over a compact subset of $\mathbb{R}^n$ to any desired accuracy up to numerical precision. Moreover, when the kernel function is continuous or bounded, convergence in RKHS norm yields point wise everywhere convergence and uniform convergence over compact subsets of $\mathbb{R}^n$. Given a Koopman operator $\mathcal{K}_F$ corresponding to the discrete dynamics, $F$, an $\epsilon > 0$, and a compact subset $\Omega$ that is invariant for the dynamics, it follows that if $\phi$ satisfies

$$\|K_F \phi - \lambda \phi\|_H \leq \epsilon, \tag{1}$$

then

$$|\phi(F^m(x)) - \lambda^m \phi(x)| \leq C\epsilon \frac{1 - \lambda^{m+1}}{1 - \lambda} \tag{2}$$

for $x \in \Omega$, where the accuracy of the resultant models depend on $\epsilon$ and $\lambda$. Significantly, as $\epsilon$ tends to zero, so does the difference $|\phi(F^m(x)) - \lambda^m \phi(x)|$ at every point in $x \in \Omega$. This everywhere convergence stands in contrast to ergodic methods, where convergence results only hold almost everywhere.

The study of dynamical systems through the Koopman formalism over RKHSs manifests as a search for functions that are close to being eigenfunctions of the Koopman operator, rather than the actual eigenfunctions. Since only a finite amount of data can be available for the study of an infinite dimensional operator, actual eigenfunctions typically cannot be computed. In fact, there is no requirement that $\mathcal{K}_F$ even has any eigenfunctions. The universality property arises in the search for Koopman "eigenfunctions," which, given a particular Koopman operator and kernel space, might not exist, since the existence of an eigenfunction depends on the Hilbert space. However, the formal equation, $\varphi(F(x)) - \lambda\varphi(x) = 0$ may still hold for some continuous function $\varphi$. If one is working over a RKHS corresponding to a universal kernel, then for any given compact set and $\varepsilon > 0$, there is a function $\tilde{\varphi} \in H$ such that $|\varphi(x) - \tilde{\varphi}(x)| < \varepsilon$ for all $x$ in that compact set, which provides for an "approximate" eigenfunction (Steinwart & Christmann, 2008). This is particularly important for DMD methods, which attempt to construct a finite rank approximation of a Koopman operator from a collection of observed snapshots. Note that obtaining approximate eigenfunctions as in equation 1 is not dissimilar to the objective of ergodic methods, where approximation of system invariants and eigenfunctions using time averages is sought. The existence of eigenfunctions depends on the selection of the Hilbert space, as will be shown in Section 4, and eigenfunctions may not be present even in the $L^2$ ergodic setting (Budišić et al., 2012).

The objective of this manuscript is to present the kernel perspective of Koopmanism as a distinct study from the ergodic perspective. In addition, this manuscript addresses several properties often assumed to hold for Koopman operators, and provides counterexamples showing that these properties do not hold in general for Koopman analysis over RKHSs. Finally, we will give a general and simplified algorithm for DMD using Koopman operators that only requires that the operators be densely defined.

## 2 Reproducing Kernel Hilbert Spaces

A Reproducing Kernel Hilbert Space (RKHS) $H$ over a set $X$ is a Hilbert space composed of functions from $X$ to $\mathbb{C}$ such that for all $x \in X$ the evaluation functional $E_x f := f(x)$ is bounded. Therefore, for all $x \in X$ there exists a function $K_x \in H$ such that $f(x) = \langle f, K_x \rangle$ for all $f \in H$. The function $K_x$ is the reproducing

kernel centered at $x$ and the function $K : X \times X \to \mathbb{C}$ defined by $K(x, y) = \langle K_y, K_x \rangle$ is the unique kernel function corresponding to $H$ (Aronszajn, 1950). Throughout most of this manuscript, the methods used will be restricted to RKHSs of real valued functions. However, for some specific examples in Section 4, it will be more convenient to employ RKHSs of complex valued functions.

Reproducing kernels can be equivalently expressed as realizations of inner products of feature space mappings in $\ell^2(\mathbb{N})$. In particular, given an orthonormal basis for a RKHS, $\{e_m(\cdot)\}_{m=1}^\infty \subset H$, the kernel function may be expressed as $K(x, y) = \sum_{m=1}^\infty e_m(x)\overline{e_m(y)}$, where $\Psi(x) := (e_1(x), e_2(x), \dots) \in \ell^2(\mathbb{N})$ is called a feature map. Equivalently, given a feature mapping $\Psi : X \to \ell^2(\mathbb{N})$, there is a RKHS whose kernel function is given as $K(x, y) = \langle \Psi(x), \Psi(y) \rangle_{\ell^2}$.

We will make repeated use of projections onto finite dimensional vector spaces arising from spans of collections of kernels centered at snapshots from a dynamical system. For a collection of centers $\{x_1, \dots, x_m\}$, the projection of a function $g \in H$ onto $\alpha = \text{span}\{K_{x_1}, \dots, K_{x_m}\}$ is given as $\arg\min_{h \in \alpha} \|h - g\|_H$, which can be resolved by expressing $h$ as $h = \sum_{i=1}^m w_i K_{x_i}$, expanding $\|\sum_{i=1}^m w_i K_{x_i} - g\|_H^2$ via inner products, and setting the derivative with respect to $w := (w_1, \dots, w_m)^T \in \mathbb{R}^m$ to zero, resulting in the weights

$$
\begin{pmatrix} w_1 \\ \vdots \\ w_m \end{pmatrix} = \begin{pmatrix} K(x_1, x_1) & \cdots & K(x_1, x_m) \\ \vdots & \ddots & \vdots \\ K(x_m, x_1) & \cdots & K(x_m, x_m) \end{pmatrix}^{-1} \begin{pmatrix} g(x_1) \\ \vdots \\ g(x_m) \end{pmatrix}
$$

The projection is then defined by $P_\alpha g := \sum_{i=1}^m w_i K_{x_i}$ where the weights $w_i$ are obtained as above.

A RKHS of real valued functions, $H$, over $\Omega \subset \mathbb{R}^n$, is said to be universal if for any compact $V \subset \Omega$, $\epsilon > 0$, and $h \in C(V)$, there is a function $\tilde{h} \in H$ such that $\|h - \tilde{h}\|_\infty < \epsilon$, where $C(V)$ denotes the set of continuous functions defined on $V$. Many commonly used kernel functions satisfy this universality property, including the Gaussian RBF kernel functions and the exponential dot product kernel functions (Steinwart & Christmann, 2008).

## 3 Koopman Operators over RKHSs

The theory of Koopman operators has been long intertwined with ergodic theory, where ergodic theoretic methods justify almost everywhere convergence claims of time averaging methods to invariants of the Koopman operator. The Birkhoff and Von Neumann ergodic theorems are posed over $L^1(\mathbb{R})$ and $L^p(\mathbb{R})$ for $p > 1$, respectively (Walters, 2000). However, the invariants for Koopman operators are not always analytic or even smooth. Hence, ergodic theorems do not give guarantees of convergence within most RKHSs, which are frequently composed of real analytic functions. Furthermore, even though ergodic theorems guarantee the existence of invariants over $L^2(\mathbb{R})$, the invariant itself is hidden behind a limiting operation via time averages (Walters, 2000). Hence, there is an expected error in the constructed invariant that stems from finiteness of data.

The objective of DMD methods is to find functions within the function space that nearly achieve eigenfunction behavior. Specifically, for a Koopman operator, $\mathcal{K}_F$, and $\epsilon > 0$, the objective is to find $\hat{\varphi} \in H$ and $\lambda \in \mathbb{C}$ for which $|\mathcal{K}_F\hat{\varphi}(x) - \lambda\hat{\varphi}(x)| < \epsilon$ for all $x$ in a given workspace. When such a function is discovered, the eigenfunction witnesses the snapshots, $x_{i+1} = F(x_i)$, as an exponential function as $\hat{\varphi}(x_{i+1}) = \lambda^i\hat{\varphi}(x_1) + \frac{1-\lambda^{i+1}}{1-\lambda} \cdot \epsilon$. If $\hat{\varphi}$ is a proper eigenfunction for $\mathcal{K}_F$, then $\epsilon$ may be taken to be zero, and $\hat{\varphi}(x_{i+1}) = \lambda^i\hat{\varphi}(x_1)$.

Ergodic methods generally yield $|\mathcal{K}_F\hat{\varphi}(x) - \lambda\hat{\varphi}(x)| < \epsilon$ only for almost all $x$ within the domain of interest. For a RKHS, the condition $|\mathcal{K}_F\hat{\varphi}(x) - \lambda\hat{\varphi}(x)| < \epsilon$ may be relaxed to $\|\mathcal{K}_F\hat{\varphi} - \lambda\hat{\varphi}\|_H < \epsilon$, since $|\mathcal{K}_F\hat{\varphi}(x) - \lambda\hat{\varphi}(x)| < C\|\mathcal{K}_F\hat{\varphi} - \lambda\hat{\varphi}\|_H < C\epsilon$, where $C > 0$ depends on the kernel function and the point $x$. If the kernel function is continuous and the domain is compact, a finite $C$ may be selected uniformly for that domain. In the special case of the Gaussian RBF kernel function and the domain being $\mathbb{R}^n$, $C$ may be taken to be 1.

If $\mathcal{K}_F$ is compact and the finite rank approximation of $\mathcal{K}_F$, which we will denote as $\hat{\mathcal{K}}_F$, is within $\epsilon$ of $\mathcal{K}_F$ with respect to the operator norm, and if $\hat{\varphi}$ is a normalized eigenfunction for $\hat{\mathcal{K}}_F$ with eigenvalue $\lambda$, then $\|\mathcal{K}_F\hat{\varphi} - \lambda\hat{\varphi}\|_H \leq \|\mathcal{K}_F\hat{\varphi} - \hat{\mathcal{K}}_F\hat{\varphi}\|_H \leq \|\mathcal{K}_F - \hat{\mathcal{K}}_F\| \leq \epsilon$. Hence, if we obtain a finite rank approximation of $\mathcal{K}_F$

that is within $\epsilon$ with respect to the operator norm, then an eigenfunction of the finite rank approximation will approximate the behavior of an eigenfunction of $\mathcal{K}_F$. This approximation is important in DMD, where the eigenfunctions are utilized to generate an approximation of the full state observable, $g_{\mathrm{id}}(x) := x$, one dimension at a time, as outlined in Section 5.

If an accurate finite rank approximation of the Koopman operator can be obtained in a RKHS, then the approximation of the overall model is accurate point-wise everywhere, and uniformly over compact sets when the RKHS consists of continuous functions. In contrast, the approximation is only accurate almost everywhere when considering Koopman operators posed over $L^2(\mathbb{R})$. Point-wise everywhere approximation is a distinctive advantage of kernel based methods that is not clear from an invocation of the kernel trick in machine learning, and requires the operator theoretic considerations introduced in this manuscript.

The need for approximation of the Koopman operator in the operator norm topology naturally leads to the question of when a Koopman operator, or more generally, a composition operator, can be compact. Compactness is a central issue for DMD as every approximation is indeed of finite rank, stemming from the observed data. In 1979, it was established in Singh & Kumar (1979) that composition operators over $L^2(\mu)$ cannot be compact when $\mu$ is non-atomic. Indeed, $L^2(\mathbb{R})$ has no compact composition operators.

Most Koopman operators over most frequently considered RKHSs are compact for a very narrow range of dynamics. In fact, most Koopman operators over these spaces are not even bounded, as will be expanded upon in Section 4.2.3. Unboundedness is an added complication for the valid implementation of DMD, addressed in Section 5, using densely defined and potentially unbounded Koopman operators. Alternatively, other classes of compact operators over RKHSs connected to dynamical systems can be leveraged for DMD procedures to give convergence guarantees (see, e.g., Rosenfeld et al. (2022)).

The remainder of this section introduces densely defined Koopman operators over RKHSs, where Lemma 1 enables the DMD algorithm introduced in Section 5. Let $H$ be a RKHS over $\mathbb{R}^n$. For a function $F : \mathbb{R}^n \to \mathbb{R}^n$ we define the Koopman Operator (sometimes called a composition operator), $\mathcal{K}_F : \mathcal{D}(\mathcal{K}_F) \to H$, as $\mathcal{K}_F g = g \circ F$ where $\mathcal{D}(\mathcal{K}_F) = \{g \in H : g \circ F \in H\}$. When $\mathcal{D}(\mathcal{K}_F)$ is dense in $H$, $\mathcal{K}_F$ is said to be densely defined. While not all densely defined Koopman operators over RKHSs are bounded, they are all closed operators.

**Lemma 1.** *Let $F : X \to X$ be the symbol for a Koopman operator over a RKHS $H$ over a set $X$. $\mathcal{K}_F : \mathcal{D}(\mathcal{K}_F) \to H$ is a closed operator.*

*Proof.* Suppose that $\{g_m\}_{m=1}^{\infty} \subset \mathcal{D}(\mathcal{K}_F)$ such that $g_m \to g \in H$ and $\mathcal{K}_F g_m \to h \in H$. To show that $\mathcal{K}_F$ is closed, we must show that $g \in \mathcal{D}(\mathcal{K}_F)$ and $\mathcal{K}_F g = h$. This amount to showing that $h = g \circ F$, by the definition of $\mathcal{D}(\mathcal{K}_F)$. Fix $x \in X$, then

$$h(x) = \langle h, K_x \rangle_H = \lim_{m \to \infty} \langle \mathcal{K}_F g_m, K_x \rangle_H = \lim_{m \to \infty} g_m(F(x))$$
$$= \lim_{m \to \infty} \langle g_m, K_{F(x)} \rangle_H = \langle g, K_{F(x)} \rangle_H = g(F(x)).$$

As $x$ was an arbitrary point in $X$, $h = g(F(x))$ and the proof is complete. $\qquad\square$

If a Koopman operator is densely defined, its adjoint is densely defined and closed. Given the kernel function centered at $x$, $\langle \mathcal{K}_F g, K_x \rangle_H = \langle g \circ F, K_x \rangle_H = g(F(x)) = \langle g, K_{F(x)} \rangle_H$. Thus, $K_x \in \mathcal{D}(\mathcal{K}_F^*)$ for all $x \in X$, and $\mathcal{K}_F^* K_x = K_{F(x)}$. Hence, each kernel function is in the domain of the adjoint of a densely defined Koopman operator, and as the span of kernel functions is dense in their RKHS, the adjoint is densely defined.

## 4 A Different Landscape for Koopman Operators

This section examines properties of Koopman operators over RKHSs. The selection of space fundamentally changes the behavior of Koopman operators over that space, where properties such as the lattice of eigenfunctions, common eigenfunctions for different discretizations, and boundedness of the operators may not hold. Each succeeding subsection provides counter examples for each of these properties for specific spaces,

and it is proven that the Gaussian RBF's native space only supports bounded Koopman operators if the discrete dynamics are affine.

Recently, kernel methods have been adapted for the study of DMD and Koopman operators, largely through the guise of extended DMD, where kernels are leveraged to simplify computations via the kernel trick. However, the adjustment from the classical study of Koopman operators through ergodic theory to that of reproducing kernel Hilbert spaces leads to significant differences in the Koopman operators and their properties. In most cases, the ergodic theorem cannot be directly applied to recover invariants of Koopman operators, since those invariants are often nonsmooth. This present section exemplifies some of the distinguishing properties of Koopman operators over RKHSs, and in some cases illustrates their limitations.

Much of the classical properties of Koopman operators in a variety of specific contexts, such as $L^2$ spaces of invariant measures and $L^1$ spaces, can be seen in Budišić et al. (2012); Kawahara (2016); Kutz et al. (2016b); Brunton & Kutz (2019); Brunton et al. (2021). Properties of the Koopman operator strongly depend on the selection of underlying vector space, and boundedness, compactness, eigenvalues, etc. change based on this selection. While Koopman operators were introduced by Koopman in 1931 in Koopman (1931) and then later picked up by the data science community in the early 2000s (e.g. (Mezić, 2005; Kutz et al., 2016b)), the study of such operators and their properties continued in earnest throughout the 20th century as composition operators (e.g. (Shapiro, 2012)). This is particularly important for RKHSs, where the specification of a bounded or densely defined Koopman operator over a particular space yields strong restrictions on the available dynamics.

### 4.1 Concerning Sampling and Discretizations

#### 4.1.1 Forward Complete Dynamics

In applications, Koopman operators enter the theory of continuous time dynamics through a discretization of the continuous time dynamical system (Bittracher et al., 2015; Mauroy & Mezić, 2016). That is, given the dynamical system $\dot{x} = f(x)$, the system is discretized through the selection of a fixed time-step, $\Delta t > 0$, as $x_{m+1} = x_m + \int_{t_m}^{t_m + \Delta t} f(x(t))dt$, where the right hand side plays the role of the discrete dynamics. However, for such a discretization to exist for arbitrarily large values of $m$, it is necessary that the dynamics be *forward complete*.

For example, consider the one dimensional dynamics, $\dot{x} = 1 + x^2$. For fixed $0 < \Delta t < \pi/2$, the corresponding discrete time dynamics are given as $x_{m+1} = \tan(\arctan(x_m) + \Delta t)$. Setting $x_m = \tan(\pi/2 - \Delta t)$, it is clear that $x_{m+1}$ is undefined. Consequently, the composition symbol, $F(x) = \tan(\arctan(x) + \Delta t)$ for the hypothetical Koopman operator, is not well defined over $\mathbb{R}^n$ for any selection of $\Delta t$.

The forward completeness assumption restricts the class of continuous dynamics on which Koopman based methods may be applied. A DMD method that circumvents this requirement, by utilizing Liouville operators and occupation kernels, may be found in Rosenfeld et al. (2022).

#### 4.1.2 Sampling and Data Science

Ergodic based methods as employed in Budišić et al. (2012); Mezić (2005); Kutz et al. (2016b;b); Takeishi et al. (2017) provide a methodology for obtaining invariants and eigenfunctions for a Koopman operator almost everywhere. That is, by selecting a continuous representative of an equivalence class in an $L^2$ space for the invariant measure, at almost every point within the domain, time averaging against that representative will converge to an invariant of the operator. However, this is a "probability 1" result, and the number of points where it may fail can potentially be uncountable. Without any external information concerning the convergence, there is no true guarantee that at a particular selected point, the time-averaged approximation will be close to the value of an actual invariant at that point. Such computational issues is precisely where the strength of kernel methods manifests.

To illustrate the kernel method, suppose that $F : \mathbb{R}^n \to \mathbb{R}^n$ is a discretization of a dynamical system with the corresponding Koopman operator, $\mathcal{K}_F : H \to H$, and $H$ is a RKHS over $\mathbb{R}^n$ consisting of continuous

functions. Suppose further that $\epsilon > 0$ and $\hat{\mathcal{K}}_F : H \to H$ is an approximation of $\mathcal{K}_F$ such that the norm difference is bounded as $\|\mathcal{K}_F - \hat{\mathcal{K}}_F\| < \epsilon$.

Suppose that $\hat{\varphi} \in H$ and is a normalized eigenfunction of $\hat{\mathcal{K}}_F$ with eigenvalue $\lambda$. The function $\hat{\varphi}$ behaves, pointwise, as an approximate eigenfunction of $\mathcal{K}_F$, since

$$|\mathcal{K}_F \hat{\varphi}(x) - \lambda \hat{\varphi}(x)| = |\mathcal{K}_F \hat{\varphi}(x) - \hat{\mathcal{K}}_F(x) \hat{\varphi}(x)|$$
$$= |\langle (\mathcal{K}_F - \hat{\mathcal{K}}_F) \hat{\varphi}, K(\cdot, x) \rangle_H| \le \|\mathcal{K}_F - \hat{\mathcal{K}}_F\| \|K(\cdot, x)\|_H \le \epsilon \cdot C$$

and $C > 0$ is a constant that depends on the kernel function and a prespecified compact domain. The compact domain may be extended to all of $\mathbb{R}^n$ in some cases, such as when the kernel function is the Gaussian RBF kernel function. Thus, it can be seen that kernel spaces and approximations that are close to the Koopman operator, in operator norm, can provide functions that behave similar to eigenfunctions of the Koopman operator. Moreover, the difference in behavior from a proper eigenfunction is governed pointwise by how close the operator approximation is in the first place.

## 4.2 Properties of the Operators

In this section, we consider a single simple example. Consider the dynamical system $\dot{x} = \begin{pmatrix} x_2 & -x_1 \end{pmatrix}^T$, which corresponds to circular dynamics in the plane. For any fixed $\theta := \Delta t$, the discretization of this system yields the linear discrete dynamics $x_{m+1} = \begin{pmatrix} \cos(\theta) & -\sin(\theta) \\ \sin(\theta) & \cos(\theta) \end{pmatrix} x_m$. That is, the discretization corresponding to a fixed time-step results in a fixed rotation of $\mathbb{R}^2$. To simplify the presentation, we use $\mathbb{C}$ as a model for $\mathbb{R}^2$, where rotation of the complex plane reduces to multiplication by a unimodular constant, $z_{m+1} = e^{i\theta} z_m$. The corresponding discrete time dynamics will be written as $F_\theta(z) := e^{i\theta} z$.

As a function space for definition of the Koopman operator, this section will consider the classical Fock space consisting of entire functions. The Fock space is used extensively in Quantum Mechanics (Hall, 2013) and it is a space where operators have been well studied (Zhu, 2012). The Fock space is given as

$$F^2(\mathbb{C}) := \left\{ f(z) = \sum_{m=0}^{\infty} a_m z^m : \sum_{m=0}^{\infty} |a_m|^2 m! < \infty \right\}.$$

The Fock space is a RKHS, with kernel function $K(z, w) = e^{\bar{w}z}$. Kernel function for the Fock space over $\mathbb{C}^n$ may be obtained through a product of single variable kernels as $K(z, w) = e^{w^*z} = e^{\bar{w}_1 z_1} \cdots e^{\bar{w}_n z_n}$.

Closely related to the Fock space is the exponential dot product kernel, $e^{x^T y}$, where for a single variable, the exponential dot product kernel's native space may be obtained by restricting the Fock space to the reals, and then taking the real part of the restricted functions. Through a conjugation of the exponential dot product kernel, the Gaussian RBF may be obtained as $K_G(x, y) = e^{-\|x\|_2^2/2} e^{x^T y} e^{-\|y\|_2^2/2} = \exp\left(-\frac{\|x-y\|_2^2}{2}\right)$, and performing the same operation on the Fock space kernel over $\mathbb{C}^n$ yields $K_G(z, w) = e^{-z^2/2} e^{w^*z} e^{-\bar{w}^2/2} = \exp\left(-\frac{(z-\bar{w})^2}{2}\right)$, which is the kernel corresponding to the complexified native space for the Gaussian radial basis function over $\mathbb{C}^n$ (cf. (Steinwart & Christmann, 2008)). This space may be expressed as

$$H_G^2(\mathbb{C}) = \left\{ g(z) e^{-z^2/2} : g \in F^2(\mathbb{C}^n) \right\},$$

and the native space corresponding to the Gaussian RBF can be obtained by taking the real parts of functions from $H_G^2$ and restricting to $\mathbb{R}^n$.

### 4.2.1 Lattice of Eigenfunctions

As presented in Budišić et al. (2012); Klus et al. (2015), the eigenfunctions of Koopman operators over $L^1(\mathbb{R})$ form a lattice. That is if $\varphi_1$ and $\varphi_2$ are two eigenfunctions for the Koopman operator, then so is $\varphi_1 \cdot \varphi_2$. For

the lattice to occur more generally, it is necessary for the product of the eigenfunctions to be a member of the underlying vector space. This closure property holds, for example, in the space of continuous functions and other Banach algebras. Hilbert spaces are not generally Banach algebras, and since it is desirable to work over Hilbert spaces for properties such as best approximations, projections, and orthonormal bases (cf. (Folland, 1999)), it is important to demonstrate that the closure property of eigenfunctions of Koopman operators does not hold in general.

Setting $\theta = \pi$, the discrete dynamics corresponding to rotation by $\pi$ in the complex plane becomes $z_{m+1} = e^{i\pi} z_m = -z_m$. That is the corresponding Koopman operator, $\mathcal{K}_{F_\pi} : F^2(\mathbb{C}) \to F^2(\mathbb{C})$, is given as $\mathcal{K}_{F_\pi} g(z) := g(-z)$. Hence, every even function is an eigenfunction for this Koopman operator with eigenvalue 1.

Any function $g \in F^2(\mathbb{C})$ exhibits a strict bound on its growth rate (cf. (Zhu, 2012)). To wit, $|g(z)| = |\langle g, K(\cdot, z) \rangle_{F^2(\mathbb{C})}| \le \|g\|_{F^2(\mathbb{C})} \|K(\cdot, z)\|_{F^2(\mathbb{C})} = \|g\|_{F^2(\mathbb{C})} e^{\frac{|z|^2}{2}}$. That is, if a function is in the Fock space then the function is of order at most 2, and if the function is of order 2 it has type at most $1/2$ (cf. (Boas, 2011)). Conversely, if an entire function is of order less than 2, it is in the Fock space, and if it is of order 2 and type less than $1/2$, then it is also in the Fock space. While functions of order 2 and type $1/2$ can be in the Fock space, it does not hold for every such function. For example, $e^{z^2/2}$ is of order 2 and type $1/2$, but is not in the Fock space.

Thus, $\varphi(z) = e^{z^2/4}$ is an eigenfunction for $\mathcal{K}_{F_\pi}$ in the Fock space. However, $\varphi \cdot \varphi = e^{z^2/2}$ is not in the Fock space, and cannot be an eigenfunction for $\mathcal{K}_{F_\pi} : F^2(\mathbb{C}) \to F^2(\mathbb{C})$. *Hence, the eigenfunctions for $\mathcal{K}_{F_\pi}$ do not form a lattice.*

### 4.2.2 Common Eigenfunctions

The intuition behind the use of Koopman operators in the study of continuous time dynamical systems is that eigenfunctions for the Koopman operators should be "close" to that of the Koopman generator for small timesteps. However, semi-groups of Koopman operators do not always share a common collection of eigenfunctions.

For example, set $\theta = \pi/2$, which yields $F_{\pi/2}(z) = iz$. In this case, the polynomial $z^4 + z^8 \in F^2(\mathbb{C})$ is an eigenfunction for the Koopman operator corresponding to $\theta = \pi/2$ with eigenvalue 1. However, $z^4 + z^8$ is not an eigenfunction for $\mathcal{K}_{F_{\pi/3}}$, as $(e^{i\pi/3} z)^4 + (e^{i\pi/3} z)^8 = e^{i4\pi/3} z^4 + e^{i2\pi/3} z^8$, and the constants cannot be factored out of the polynomial as an eigenvalue.

Hence, since each Koopman operator obtained through a fixed time-step may produce a different collection of eigenfunctions, there is no way to distinguish which, if any, should correspond to eigenfunctions of the Koopman generator.

### 4.2.3 Boundedness of Koopman Operators

Throughout the literature, it is frequently assumed that Koopman operators are bounded. This assumption manifests as an unrestricted selection of observables in the study of the Koopman operator. When a Koopman operator is a densely defined operator whose domain is the entire Hilbert space, it is also closed. Hence, by the closed graph theorem (cf. (Folland, 1999, Theorem 5.12)), such an operator must be bounded. Furthermore, the collection of finite rank operators is dense in the collection of bounded operators over a Hilbert space in the strong operator topology (SOT) (cf. (Pedersen, 2012, Paragraph 4.6.2)). Convergence in SOT was independently studied in the work Korda & Mezić (2018), where the DMD routine was demonstrated to converge to a bounded Koopman operator in SOT.

As mentioned in Korda & Mezić (2018), SOT convergence does not in general lead to convergence of the eigenvalues. To preserve spectral convergence, the finite rank approximations produced by DMD algorithms need to converge to Koopman operators in the operator norm topology. The most direct approach, and one that leads to good pointwise estimates of eigenfunctions, is through the use of compact Koopman operators. However, it isn't immediately clear when one can expect a continuous dynamical system to yield a compact Koopman operator through discretization. For example, the Koopman operator corresponding to

discretization of the continuous time system $\dot{x} = 0$ is the identity operator, $I$, for any fixed time step, and $I$ is not compact over any infinite dimensional Hilbert space.

In addition, for any given RKHS, the collection of bounded Koopman operators is very small. It was demonstrated in Carswell et al. (2003) that a Koopman operator over the Fock space is bounded only when the corresponding discrete dynamics are *affine*. It follows that the same result holds over the exponential dot product kernel's native space.

It may perhaps be less obvious that this result extends to the Gaussian RBF's native space, and a proof of this fact is given below. As far as the authors are aware, this is the first time this result has appeared in the literature, and it demonstrates that even for popular selections of RKHSs, the collection of bounded Koopman operators is small.

**Lemma 2.** *If $\mathcal{K}_F$ is a bounded operator over the Gaussian RBF's native space, then $F$ is a real analytic vector valued function over $\mathbb{R}^n$.*

*Proof.* If $\mathcal{K}_F$ is bounded, then $\mathcal{K}_F K_y(x) = K_y(F(x)) = \exp(-\|F(x) - y\|_2^2)$ is in the RBF's native space for each $y \in \mathbb{R}^n$. Since every function in the RBF's native space is real analytic, so is $K_y(F(x))$, and thus, the logarithm, $-\|F(x) - y\|_2^2 = -\|F(x)\|_2^2 + 2y^T F(x) - \|y\|_2^2$ is real analytic. This holds if $y = 0$, and hence $\|F(x)\|_2^2$ is real analytic. Thus, for every $y$, the quantity $y^T F(x)$ is real analytic. That each component of $F(x)$ is real analytic follows from the selection of $y$ as the cardinal basis elements of $\mathbb{R}^n$, and this completes the proof. $\square$

**Lemma 3.** *If $F$ is a real analytic vector valued function that yields a bounded Koopman operator, $\mathcal{K}_F$, over the Gaussian RBF's native space, then its extension to an entire function, $F : \mathbb{C}^n \to \mathbb{C}^n$ induces a bounded operator over $H_G(\mathbb{C}^n)$.*

*Proof.* Since an entire function on $\mathbb{C}^n$ is uniquely determined by its restriction to $\mathbb{R}^n$, it follows that the span of the complex valued Gaussian RBFs with centers in $\mathbb{R}^n$ is dense in $H_G$. Moreover, to demonstrate that $\mathcal{K}_F$ is bounded, it suffices to show that there is a constant $C$ such that

$$C^2 K_G(z, w) - K_G(F(z), F(w)) \tag{3}$$

is a positive kernel. By the above remark, it suffices to show this for real $x, y \in \mathbb{R}^n$, but then this is equivalent to the statement that $\mathcal{K}_F$ is bounded over the Gaussian RBF's native space over $\mathbb{R}^n$. $\square$

**Theorem 1.** *If $F : \mathbb{C}^n \to \mathbb{C}^n$ is an entire function, and $\mathcal{K}_F$ is bounded on $H_G$, then $F(z) = Az + b$ for a matrix $A \in \mathbb{C}^{n \times n}$ and vector $b \in \mathbb{C}^n$.*

*Proof.* If $\mathcal{K}_F$ is bounded, then it has a bounded adjoint, $\mathcal{K}_F^*$, which acts on the complex Gaussian as $\mathcal{K}_F^* K_G(\cdot, z) = K_G(\cdot, F(z))$. In particular, there is a constant $C > 0$ such that $\|K_G(\cdot, F(z))\|_{H_G}^2 \leq C^2 \|K_G(\cdot, z)\|_{H_G}^2$. Noting the identity $\|K_G(\cdot, z)\|_{H_G}^2 = \exp\left(2 \sum_{j=1}^n (\Im z_j)^2\right)$ and taking the logarithm, it follows that

$$\sum_{j=1}^n (\Im F_j(z))^2 \leq \log(C^2) + \sum_{j=1}^n (\Im z_j)^2 \leq \log(C^2) + \|z\|_2^2. \tag{4}$$

From this inequality, it follows that for each coordinate $j = 1, \ldots, n$, the harmonic function $v_j(z) = \Im F_j(z)$ has linear growth. That is, there is a constant $\tilde{C}$ so that $|v_j(z)| \leq \tilde{C}(1 + \|z\|_2)$ for all $z \in \mathbb{C}^n$. It follows (e.g. from the standard Cauchy estimates) that $v_j(z) = v_j(x + iy)$ must be an affine linear function of $x$ and $y$, and therefore, so must its harmonic conjugate $u_j(z)$, and we conclude that $F(z) = Az + b$. $\square$

**Corollary 1.** *If $F$ is a real entire vector valued function, and $\mathcal{K}_F$ is bounded on the Gaussian RBF's native space over $\mathbb{R}^n$, then $F$ is affine.*

Hence, for the most commonly used kernel function in machine learning, the collection of bounded (and hence compact) Koopman operators over its native space is restricted to only those Koopman operators corresponding to affine dynamics. Each selection of RKHS and kernel function will yield a correspondingly

small collection of bounded Koopman operators. It should be noted that Koopman operators were completely classified for over the classical sampling space, the Paley-Wiener space (Chacón & Giménez, 2007), as also being those that correspond to affine dynamics, and it is a simple exercise to show that the native space for the polynomial kernel also only admits bounded Koopman operator when the dynamics are affine.

Consequently, in most practical respects Koopman operators over RKHSs should not be assumed to be bounded, and certainly not compact.

## 5 Dynamic Mode Decomposition with Koopman Operators over RKHSs

As a product of its genesis in the machine learning community, many DMD procedures appeal to feature space, and this continues to hold in the current implementations of kernel-based extended DMD (Williams et al., 2015b), which casts the snapshots from a finite dimensional nonlinear system into an infinite feature space. The direct involvement of the feature space in the estimation of the Koopman operator leads to rather complicated numerical machinery. To avoid directly computing the infinite dimensional vectors that result, an involved collection of linear algebra techniques are leveraged to extract the Koopman modes. Here it is shown that this process may be simplified and that a procedure that directly involves the kernel functions centered at the snapshots simplifies the design of DMD algorithms. This approach keeps to the spirit of the "kernel trick," where feature vectors are never directly evaluated and only accessed through evaluations of the kernel function itself.

Recall that in a real valued Hilbert space, the projection of a function $g$ onto a collection of linearly independent basis functions, $u_1, \ldots, u_M$, is a linear combination of those functions, $Pg = \sum_{j=1}^{M} w_j u_j$ where the weights $w_j$ may be determined via

$$\begin{pmatrix} w_1 \\ \vdots \\ w_M \end{pmatrix} = \begin{pmatrix} \langle u_1, u_1 \rangle_H & \cdots & \langle u_1, u_M \rangle_H \\ \vdots & \ddots & \vdots \\ \langle u_M, u_1 \rangle_H & \cdots & \langle u_M, u_M \rangle_H \end{pmatrix}^{-1} \begin{pmatrix} \langle g, u_1 \rangle_H \\ \vdots \\ \langle g, u_M \rangle_H \end{pmatrix}.$$

Let $E$ be a linear transformation between two finite dimensional vector spaces $V$ and $W$. Suppose that $\alpha = \langle \alpha_1, \ldots, \alpha_N \rangle$ is an ordered basis for $V$, and suppose that $\beta = \langle \beta_1, \ldots, \beta_M \rangle$ is an ordered basis for $W$. The matrix representation of the linear transformation, $E$, with respect to the ordered bases $\alpha$ and $\beta$ is denoted as $[E]_\alpha^\beta$, where the $j$-th column of this matrix is the weights of the vector $E_{\alpha_j}$ corresponding to the ordered basis $\beta$.

Throughout this algorithm, a Koopman operator will be assumed to be densely defined, as Section 4 demonstrated that most Koopman operators cannot be expected to be bounded or compact. An additional assumption will be made that the kernel functions themselves reside in the domain of the Koopman operator. It should be noted that since the kernels are always in the domain of the adjoint of the Koopman operator (see Section 3), a finite rank representation of the adjoint of the Koopman operator may thus be derived without assuming that the kernels are in the domain of the Koopman operator.

For the sake of the derivation, it is also assumed that the Koopman operator is diagonalizable, which is not generally expected to be true. However, the finite rank representations leveraged in this manuscript are almost always diagonalizable, since the set of non-diagonalizable matrices are of measure zero in the collection of all matrices. Moreover, for periodic data sets, the adjoint of the Koopman operator will be invariant on the span of the collection of kernel functions centered at the snapshots, and thus, the finite rank representations will be explicitly the adjoint of the Koopman operator on that subspace, which supports the assumption of the availability of eigendecompositions for the Koopman operator in the periodic or quasiperiodic settings.

For a given collection of snapshots $\{x_1, x_2, ..., x_m\}$[1], the goal is to determine a finite rank representation of $\mathcal{K}_F$ that is derived from the kernel functions centered at the snapshots. To express a finite rank representation,

---

[1]While availability of a time series of snapshots $\{x_1, x_2, ..., x_m\}$ such that $x_{i+1} = F(x_i)$ is a more typical use case, the developed method does not require such a time series. It can also be implemented using arbitrary snapshots $\{x_1, x_2, ..., x_m\}$ and $\{y_1, y_2, ..., y_m\}$ provided $y_i = F(x_i)$.

the ordered basis $\alpha = \{k_{x_1}, ..., k_{x_{m-1}}\}$ is leveraged. In particular, if $P_\alpha$ is the projection on to span($\alpha$), the operator $P_\alpha \mathcal{K}_F$ maps span($\alpha$) to itself, which enables the discussion of eigenfunctions and eigenvalues of $\mathcal{K}_F$ using only functions in span($\alpha$).

Suppose that given a function $g \in \text{span}\,\alpha$, with coefficients $a_1, ..., a_{m-1}$, the function $P_\alpha \mathcal{K}_F g$ can be expressed in the basis $\alpha$ using the coefficients $b_1, ..., b_{m-1}$. As a result, the operator $P_\alpha \mathcal{K}_F$ can be represented using a matrix that maps the vector $(a_1, ..., a_{m-1})^T$ to the vector $(b_1, ..., b_{m-1})^T$. In the following development, the finite rank representation of $P_\alpha \mathcal{K}_F$, expressed in a matrix form, is denoted by $[P_\alpha \mathcal{K}_F]_\alpha^\alpha$, where the notation $[\cdot]_\alpha^\alpha$ indicates that both the domain and range of $P_\alpha \mathcal{K}_F$ is restricted to span($\alpha$).

The $i$th column of $[P_\alpha \mathcal{K}_F]_\alpha^\alpha$ may be determined through the examination of the action of the operator $P_\alpha \mathcal{K}_F$ on the basis function $k_{x_i}$, for $i = 1, \ldots, m-1$ as

$$
\begin{pmatrix} \langle k_{x_1}, k_{x_1} \rangle_H & \cdots & \langle k_{x_1}, k_{x_{m-1}} \rangle_H \\ \vdots & \ddots & \vdots \\ \langle k_{x_{m-1}}, k_{x_1} \rangle_H & \cdots & \langle k_{x_{m-1}}, k_{x_{m-1}} \rangle_H \end{pmatrix}^{-1} \begin{pmatrix} \langle P_\alpha \mathcal{K}_F k_{x_i}, k_{x_1} \rangle_H \\ \vdots \\ \langle P_\alpha \mathcal{K}_F k_{x_i}, k_{x_{m-1}} \rangle_H \end{pmatrix}
$$

Therefore, using the fact that $\mathcal{K}_F k_{x_i}(x) = k_{x_i}(F(x))$, the matrix $[P_\alpha \mathcal{K}_F]_\alpha^\alpha$ may be written as

$$
[P_\alpha \mathcal{K}_F]_\alpha^\alpha = \begin{pmatrix} K(x_1, x_1) & \cdots & K(x_1, x_{m-1}) \\ \vdots & \ddots & \vdots \\ K(x_{m-1}, x_1) & \cdots & K(x_{m-1}, x_{m-1}) \end{pmatrix}^{-1} \begin{pmatrix} K(x_2, x_1) & \cdots & K(x_2, x_{m-1}) \\ \vdots & \ddots & \vdots \\ K(x_m, x_1) & \cdots & K(x_m, x_{m-1}) \end{pmatrix}.
$$

It should be noted that this is precisely the pair of matrices examined in Williams et al. (2015b) after the use of a truncated SVD. In numerical implementations of the developed method, if the Gram matrix being inverted above is not invertible, then it is regularized by adding $\epsilon I_{m-1}$ to it, where $\epsilon > 0$ is a user-selected regularization coefficient and $I_{m-1}$ is an $m - 1 \times m - 1$ identity matrix.

The objective of DMD is to use the finite rank representation determined above to create a data driven model of the dynamical system. This makes use of a fundamental property of eigenfunctions of the Koopman operator. In particular, suppose that $\varphi$ is an eigenfunction of $\mathcal{K}_F$ with eigenvalue $\lambda$. Evaluating the eigenfunction at a snapshot reveals $\varphi(x_{i+1}) = \varphi(F(x_i)) = \mathcal{K}_F \varphi(x_i) = \lambda \varphi(x_i)$. Hence, $\varphi(x_{i+1}) = \lambda^i \varphi(x_1)$. Now suppose that $\{\varphi_j\}_{j=1}^\infty$ is an eigenbasis for a diagonalizable Koopman operator, $\mathcal{K}_F$, corresponding to the eigenvalues $\{\lambda_j\}_{j=1}^\infty$. For a state $x \in \mathbb{R}^n$, let $(x)_d$ be the $d$-th component of $x$ for $d = 1, \ldots, n$. If it is assumed that the mapping $x \mapsto (x)_d$ is in the RKHS (as it is when $H$ is the native space for the exponential dot product space (Steinwart & Christmann, 2008)), then it may be expressed as $(x)_d = \lim_{M \to \infty} \sum_{j=1}^M (\xi_{j,M})_d \varphi_j(x)$ for some coefficients $\{(\xi_{j,M})_d\}_{j=1}^\infty$. Note that since the Koopman operator is not generally a normal operator, $\{\varphi_i\}_{i=1}^\infty$ is not expected to be an orthonormal basis, and hence, there may be nonzero influences between the coefficients obtained by projection and this is expressed by the additional index $M$ in $\xi_{j,M}$. This means that a series representation of the decomposition as expressed in Kawahara (2016); Brunton & Kutz (2019) is not always possible. *Hence, Koopman modes are not fixed quantities unless there is an orthonormal basis of eigenfunctions for the Koopman operator.* By stacking each $(x)_d$, the full state observable $g_{\text{id}}$, given by $g_{\text{id}}(x) = x$, is expressed as

$$
g_{\text{id}}(x) = \lim_{M \to \infty} \sum_{j=1}^M \xi_{j,M} \varphi_j(x). \tag{5}
$$

Hence, each snapshot may be reconstructed as

$$
x_{i+1} = \lim_{M \to \infty} \sum_{j=1}^M \xi_{j,M} \lambda_j^i \varphi_j(x_1). \tag{6}
$$

Since the Koopman operator is approximated here by a finite rank representation, perfect reproduction of $g_{\text{id}}$ through a series of eigenfunctions is not possible. Instead, eigenfunctions determined through the finite

rank representation are used to construct the approximation of $g_{\mathrm{id}}$. In particular, the matrix $[P_\alpha \mathcal{K}_F]^\alpha_\alpha$ is the matrix representation of $P_\alpha \mathcal{K}_F$. If $v_j$ is an eigenvector for the matrix $[P_\alpha \mathcal{K}_F]^\alpha_\alpha$ with eigenvalue $\lambda_j$, then

$$P_\alpha \mathcal{K}_F \left( \sum_{i=1}^{m-1} (v_j)_i K(x, x_i) \right) = \begin{pmatrix} K(x, x_1) \\ \vdots \\ K(x, x_{m-1}) \end{pmatrix}^T [P_\alpha \mathcal{K}_F]^\alpha_\alpha v_j = \lambda_j \sum_{i=1}^{m-1} (v_j)_i K(x, x_i).$$

That is, $\sum_{i=1}^{m-1} (v_j)_i K(x, x_i)$ is an eigenfunction of $P_\alpha \mathcal{K}_F$. The corresponding normalized eigenfunction is denoted by

$$\hat{\varphi}_j(x) := \frac{1}{\sqrt{v_j^\dagger G v_j}} \sum_{i=1}^{m-1} (v_j)_i K(x, x_i), \tag{7}$$

where $G = (K(x_i, x_\ell))_{i,\ell=1}^{m-1}$ is the Gram matrix associated with the snapshots and the kernel function and $(\cdot)^\dagger$ denotes the conjugate transpose.

Using a finite rank representation of equation 5, it is easy to see that the $d$-th row of the matrix $\hat{\xi} := \begin{pmatrix} \hat{\xi}_1 & \dots & \hat{\xi}_{m-1} \end{pmatrix}$ of Koopman modes is comprised of the components of $(x)_d$ along the (non-orthogonal) directions $\hat{\varphi}_j$. That is, $g_{\mathrm{id}}(x_i) = x_i = \sum_{j=1}^{m-1} \xi_j \hat{\varphi}_j(x_i)$, which yields $\hat{\xi} = X (V^T G)^{-1}$, where $X := \begin{pmatrix} x_1 & \dots & x_{m-1} \end{pmatrix}$ is the data matrix and

$$V := \begin{pmatrix} \frac{v_1}{\sqrt{v_1^\dagger G v_1}} & \cdots & \frac{v_{m-1}}{\sqrt{v_{m-1}^\dagger G v_{m-1}}} \end{pmatrix}$$

is the matrix of normalized eigenvectors of $[P_\alpha \mathcal{K}_F]^\alpha_\alpha$.

Using the approximate eigenfunctions, $\hat{\varphi}_j$, a data driven model for the system is obtained as

$$x_{i+1} \approx \sum_{j=1}^{m-1} \hat{\xi}_j \lambda_j^i \hat{\varphi}_j(x_1). \tag{8}$$

---

**Algorithm 1:** Pseudocode for the novel kernel perspective based DMD algorithm. Upon obtaining the Koopman modes, the approximate eigenfunctions, and the eigenvalues, equation 8 is used to compute $x_{i+1}$.

---

**Input:** Snapshots, $X = \{x_1, x_2, ..., x_m\}$
**Input:** Kernel function $K : \mathbb{R}^n \times \mathbb{R}^n \to \mathbb{R}$ of an RKHS

Step 1 Compute the Gram matrix, $G = (K(x_i, x_\ell))_{i,\ell=1}^{m-1}$

Step 1.5 If $G$ is ill-conditioned, then set $G = G + \epsilon I_{m-1}$

Step 2 Compute the interaction matrix, $\mathcal{B} = (K(x_i, x_\ell))_{i=2,\ell=1}^{i=m,\ell=m-1}$

Step 3 Compute the finite rank representation of the Koopman operator, $[P_\alpha \mathcal{K}_F]^\alpha_\alpha = G^{-1} \mathcal{B}$

Step 4 Compute eigenvalues, $\lambda_j$, and eigenvectors, $v_j$, of $[P_\alpha \mathcal{K}_F]^\alpha_\alpha$

Step 5 Compute the matrix of Koopman modes, $\hat{\xi} = X (V^T G)^{-1}$

Step 6 Compute approximate eigenfunctions, equation 7

**Output:** Koopman modes, $\hat{\xi}_j$ for $j = 1, \dots, m-1$
**Output:** Approximate eigenfunctions, $\hat{\varphi}_j(x)$ for $j = 1, \dots, m-1$
**Output:** Eigenvalues $\lambda_j$ for $j = 1, \dots, m-1$

---

# 6    Vector Valued Considerations

The attentive reader will notice that the ultimate objective of DMD is to achieve an approximation or decomposition of the function $g_{\text{id}}(x) := x$, the full state observable. However, the full state observable is $\mathbb{R}^n$ valued, whereas the RKHSs in question consist of scalar valued functions. Consequently, the coefficients that are determined to approximate the full state observable are vector valued. Up to now, the methods have simply separated the individual components of $g_{\text{id}}$ and established an approximation for each of them separately. The weights for these approximations are stacked to make the vector valued coefficients.

This section shows how the vector valued coefficients arise naturally from a projection onto vector valued functions in a vector valued RKHS. With the right selection of kernel operator, this projection operation reduces to the setting of Section 5.

If the goal is to decompose the full state observable all at once, then one must appeal to vector valued RKHSs, which produce an operator valued kernel function. Given a vector valued RKHS, $H$, consisting of functions that map a set $X$ to a Hilbert space $\mathcal{Y}$, there is for each $x \in X$ and $\nu \in \mathcal{Y}$ a function $K_{x,\nu} \in H$ such that $\langle g, K_{x,\nu} \rangle_H = \langle g(x), \nu \rangle_{\mathcal{Y}}$. The mapping $\nu \to K_{x,\nu}$ is linear, and hence, the kernels are expressed as $K_x \nu$ where $K_x : \mathcal{Y} \to H$ is a bounded operator. In this case, for each $g \in H$, $K_x^* g = g(x)$. The operator valued kernel associated with $H$ is given as $K(x,y) := K_x^* K_y : \mathcal{Y} \times \mathcal{Y} \to H$, and note that when $\mathcal{Y} = \mathbb{R}^n$, this operator valued kernel is a matrix whose entries are scalar valued functions.

When a vector valued RKHS, $H$, is obtained from a scalar valued RKHS, $\tilde{H}$ with kernel $k(x,y)$, as $H = \tilde{H}^n$ where the inner product of two elements, $g = (g_1, \ldots, g_n)^T$ and $h = (h_1, \ldots, h_n)^T$, from $H$ is given as $\langle g, h \rangle_H = \sum_{j=1}^n \langle g_j, h_j \rangle_{\tilde{H}}$, then the operator valued kernel is given as $k(x,y) I_n$, which has many convenient properties for computation. More details concerning vector valued RKHSs may be found in Carmeli et al. (2010); Agler & McCarthy (2023).

Here the Koopman operator is defined just as it would be for a scalar valued space: $\mathcal{K}_F g = g \circ F$, and the difference here is that $g$ is vector valued. Similar relationships between the Koopman operator and the kernel functions still hold, where $\langle \mathcal{K}_F g, K_x \nu \rangle_H = \langle g(F(x)), \nu \rangle_{\mathcal{Y}} = \langle g, K_{F(x)} \nu \rangle_H$, hence $\mathcal{K}_F^* K_x = K_{F(x)}$.

Now given a collection of snapshots, $\{x_1, \ldots, x_M\}$, the associated operator valued kernels are $K_{x_1}, \ldots K_{x_M}$. For decomposition of the full state observable, $g_{id}(x) = x$, one needs to interface with the Hilbert space and construct a finite rank approximation of $\mathcal{K}_F$ as was done in the prior section. This means that a subspace within $H$ needs to be constructed, say spanned by a collection of basis functions $\alpha$, so one can write $\tilde{\mathcal{K}}_F = P_\alpha \mathcal{K}_F P_\alpha$.

The selection of the basis $\alpha$ is less constrained than in the scalar valued case, only the kernels centered at the snapshots were considered. In this case, not only must the snapshots be considered, but also complete freedom of the choice of $\nu \in \mathbb{R}^n$ to select each $K_{x,\nu} = K_x \nu$. A natural choice is to select $\nu$ from the standard basis of $\mathbb{R}^n$. This leads to $\alpha = (K_{x_0,e_1}, \ldots, K_{x_{M-1},e_1}, K_{x_0,e_2}, \ldots, K_{x_{M-1},e_2}, \ldots, K_{x_0,e_n}, \ldots, K_{x_{M-1},e_n})$, which is a large basis for problems such as the cylinder flow example, where $n$ is of the order $80,000$. Consequently, the gram matrix corresponding to this basis is a square matrix of dimension $M \cdot n$.

So for arbitrary operator valued kernels, the DMD algorithm presented in this paper would have to invert an $M \cdot n$ dimensional matrix, and the computation time would scale with the dimension of the space as $O((M \cdot n)^3)$ using standard inversion algorithms. This scaling would make benchmarks such as the cylinder flow example completely infeasible.

Computational feasibility can be achieved through the judicious selection of the kernel operator. Here we leveraged a kernel operator of the form $K(x,y) = k(x,y) I_n$ with $k$ a scalar valued kernel function. In this case, $\langle K_x \nu, K_y \omega \rangle_H = k(x,y) \langle \nu, \omega \rangle_{\mathbb{R}^n}$ and the two functions $K_x \nu$ and $K_y \omega$ are orthogonal when $\nu$ and $\omega$ are orthogonal in $\mathbb{R}^n$. This means that the Gram matrix composed of the $\alpha$ given above is going to be a block diagonal matrix with each block being the Gram matrix corresponding to the scalar valued kernel. Thus only one matrix of size $M$ must be inverted, and each dimension treated individually. Significantly, the inversion problem no longer scales with the dimension.

Alternatively, if we instead use a different scalar valued space for each dimension, we would get $n$ different Gram matrices to invert. The complexity would scale with the dimension, but linearly rather than cubicly.

Formally, suppose that $k(x, y)$ corresponds to a scalar valued RKHS, $\tilde{H}$, and let $H$ be an $\mathbb{R}^n$ valued RKHS such that if $g \in H$, then $g = (g_1, \ldots, g_n)^T$ where $g_i \in \tilde{H}$ for each $i = 1, \ldots, n$ equipped with the inner product $\langle g, h \rangle_H = \sum_{i=1}^n \langle g_i, h_i \rangle_{\tilde{H}}$. That $H$ is a vector valued RKHS follows since if $v \in \mathbb{R}^n$ and $x \in X$ then $|\langle g(x), v \rangle_{\mathbb{R}^n}| \leq \|g(x)\|_{\mathbb{R}^n} \|v\|_{\mathbb{R}^n} = \sqrt{\sum_{i=1}^\infty |\langle g_i, k(\cdot, x) \rangle_{\tilde{H}}|^2} \|v\|_{\mathbb{R}^n} \leq \sqrt{k(x,x)} \sqrt{\sum_{i=1}^\infty \|g_i\|_{\tilde{H}}^2} \|v\|_{\mathbb{R}^n} = \sqrt{k(x,x)} \|g\|_H \|v\|_{\mathbb{R}^n}$, hence the functional $g \mapsto \langle g(x), v \rangle_{\mathbb{R}^n}$ is bounded. In this setting, $K(x, y) = \text{diag}(k(x, y), \ldots, k(x, y))$, and $\langle K_x e_i, K_y e_j \rangle_{\mathbb{R}^n} = 0$ if $i \neq j$.

If $\{x_1, \ldots, x_N\}$ is a collection of snapshots such that $F(x_i) = x_{i+1}$, then a finite rank approximation of $\mathcal{K}_F$ may be constructed by examining the image of the operator on $K_{x_i, e_j}$ for $i = 1, \ldots, N$ and $j = 1, \ldots, n$, and then projecting back onto the span of these kernels. The projection operation requires the computation of the Gram matrix for the basis $\{K_{x_i, e_j}\}$, which is a block diagonal matrix, where each block corresponds to a selection of dimension through $e_j$. Thus, if $\tilde{G} = (k(x_s, x_\ell))_{s, \ell = 1}^N$ the gram matrix manifests as

$$G = \begin{pmatrix} \tilde{G} & & \\ & \ddots & \\ & & \tilde{G} \end{pmatrix}.$$

Similarly, the interaction matrix is a block diagonal. Consequently, the vector of weights corresponding to each kernel function is composed of $n$ smaller vectors of length $N - 1$, each one corresponding to a different dimension. Hence, each dimension may be treated independently. The eigenfunctions for this finite rank representation of the Koopman operator are then composed of $n$ identical collections of $N - 1$ functions, differing only in the dimension they are supported in. Let $\tilde{\varphi}_1, \ldots, \tilde{\varphi}_N \in \tilde{H}$ be these scalar valued functions, and write $\varphi_{s,i} := \tilde{\varphi}_s e_i$ be the corresponding vector valued eigenfunction.

If the full state observable is projected onto this collection of eigenfunctions as $g_{\text{id}}(x) \approx \sum_{i=1}^n \sum_{s=1}^N w_{s,i} \phi_{s,i}(x) = \sum_{s=1}^N \tilde{\phi}_s(x) \left( \sum_{i=1}^n w_{s,i} e_i \right)$. Here, $\sum_{i=1}^n w_{s,i} e_i = \xi_s$, where $\xi_s$ is the Koopman mode from the previous section.

## 7 Numerical Examples

### 7.1 Periodic flow around a cylinder

The website (Kutz et al., 2016a) accompanying the textbook Kutz et al. (2016b) provides several data sets that serve as benchmarks for spectral decomposition approaches to nonlinear modeling, which have been released to the public through their website. This section utilizes the cylinder flow data set to demonstrate the results of the developed DMD method. The cylinder flow example is numerically generated, and the data provided corresponds to a planar steady state flow of the system. The data set consists of 151 snapshots, containing values of the vorticity of the flow at several mesh points in a plane, recorded every 0.02 seconds. In this demonstration, snapshots 1 through 30 are used as the input data, and snapshots 2 through 31 are used as output data, assuming that the $i$th and $(i+1)$th snapshots satisfy $x_{i+1} = F(x_i)$ for some unknown nonlinear function $F$. The snapshots are normalized so that the largest 2-norm among all snapshots is 1.

The Koopman modes, eigenvalues, and eigenfunctions are then computed using the developed technique and snapshots 2 through 31 are reconstructed using equation 8. DMD is implemented using the exponential dot product kernel, $K(x, y) = \exp(\frac{1}{\mu} x^T y)$ (with $\mu = 1$), and the Gaussian RBF kernel, $K(x, y) = \exp\left(-\frac{1}{\mu} \|x - y\|_2^2\right)$ (with $\mu = 1$).

To further demonstrate the accuracy of the obtained finite-dimensional representation of the Koopman operator, snapshots 32 through 151 are *predicted* from snapshot 1 using equation 8.

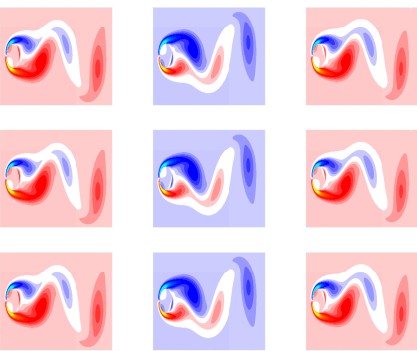

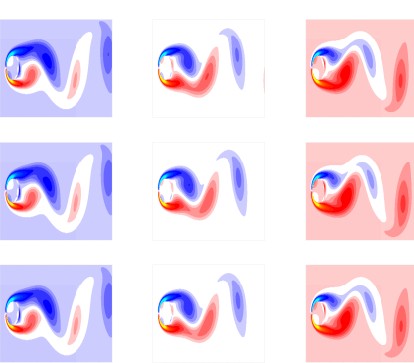

(a) This figure presents the reconstruction of the vorticity field at the same time points using two different kernel functions. The left column presents the reconstruction of the initial state, $x_1$. The middle column shows the reconstructions of the state, $x_{15}$, and the right column corresponds to the reconstruction of the state $x_{30}$.

(b) This figure presents prediction of the vorticity field at the same time points using two different kernels. The left column presents the prediction of, $x_{51}$. The middle column shows the reconstructions of the state, $x_{101}$, and the right column corresponds to the reconstruction of the state $x_{151}$.

Figure 1: Reconstruction and prediction of the vorticity of a fluid flow past a cylinder using two different kernel functions. The first rows contains the ground truth, the second rows leverages the Gaussian RBF kernel, and the third row uses the exponential dot product kernel.

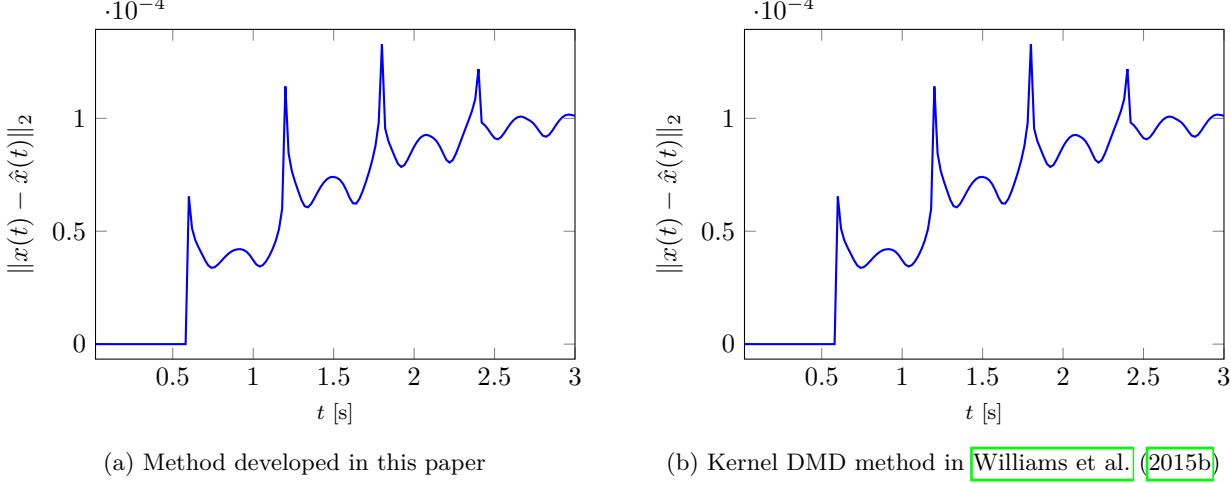

(a) Method developed in this paper

(b) Kernel DMD method in Williams et al. (2015b)

Figure 2: For the periodic flow example, this figure shows the 2-norm of the error between the true snapshots and the snapshots generated using DMD. The results for the Gaussian RBF kernel and the exponential dot product kernel are identical.

## 7.2 Turbulent flow

Another Numerical example uses a data set that accompanies Jakob et al. (2020), which consist of flow simulations that range from laminar flow configurations to turbulent flow configurations. The data can be accessed from their website Jakob et al. (2021). The flow data used in this paper (id number 7999) corresponds to 2-dimensional turbulent flow with a Reynolds number of 4092.2162409351754 and a Kinematic viscosity of $5.4501926528485274 \exp{-05}$. The data consist of 1001 snapshots, each snapshot contains flow velocities in the horizontal and vertical directions measured on a regular grid of size $512 \times 512$ in two dimensions with domain $[0,1] \times [0,1]$. The snapshots are separated by 0.01 seconds. In this demonstration, snapshots 1 through 300 are used as the input data, and snapshots 2 through 301 are used as output data,

assuming that the $i$th and $(i+1)$th snapshots satisfy $x_{i+1} = F(x_i)$ for some unknown nonlinear function $F$. The snapshots are normalized so that the largest 2-norm among all snapshots is 1.

The Koopman modes, eigenvalues, and eigenfunctions are then computed using the developed technique and snapshots 2 through 301 are reconstructed from snapshot 1 using equation 8. DMD is implemented using the Gaussian RBF kernel, $K(x,y) = \exp\left(-\frac{1}{\mu}\|x - y\|_2^2\right)$ (with $\mu = 0.75$ and $\epsilon = 0.0001$). The baseline method from Williams et al. (2015b) is implemented using the same kernel with $\mu = 0.000518$. The predictive model in equation 8 is also used to generate 700 additional snapshots.

### 7.3 Discussion

The ability of the developed DMD technique to reconstruct the training data from the same initial condition is apparent from figures 1a, 2, and 3 both for the periodic flow, where the relative reconstruction error are of the order of $1e - 4$, and the turbulent flow, where the relative reconstruction errors are of the order of $1e - 2$. As shown in 1a, both of the reproducing kernels used yield similar reconstruction results.

In the case of the periodic flow, as shown in figures 1b and 2, given the first 31 snapshots, the developed DMD technique is able to predict the remaining 120 snapshots, with a relative prediction error of order $1e - 8$, *without the knowledge of the underlying physics, $F$.* Furthermore, as expected, the performance of the developed method is nearly identical to the baseline Kernel DMD method developed in Williams et al. (2015b).

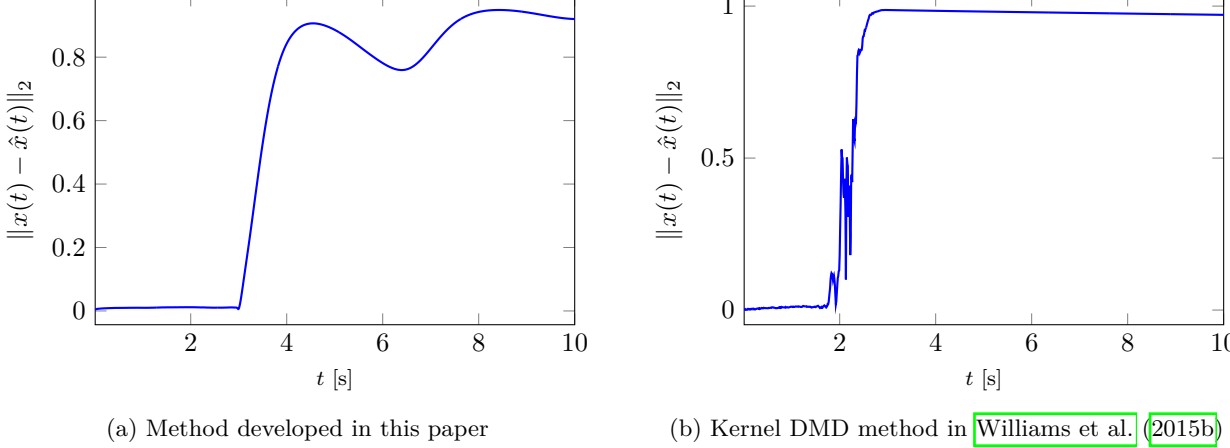

(a) Method developed in this paper        (b) Kernel DMD method in Williams et al. (2015b)

Figure 3: For the turbulent flow example, this figure shows the 2-norm of the error between the true snapshots and the snapshots generated using DMD.

In the case of the turbulent flow, as shown in 3, the relative prediction errors for the method developed in this paper and the baseline kernel DMD method developed in Williams et al. (2015b) quickly increase to 1. Since the data are normalized so that the largest 2-norm among all snapshots is 1, a relative prediction error of 1 signifies a very poor match between the predicted state and the true state. Such performance degradation is expected for turbulent flows since the underlying assumption that the $i$th and $(i + 1)$th snapshots satisfy $x_{i+1} = F(x_i)$ for some unknown nonlinear function $F$ is unlikely to generalize outside the training data when the flow is turbulent. From figure 3, the method developed in this paper appears to perform slightly better than the baseline. This performance improvement can be attributed to the explicit regularization of the Gram matrix in Step 1.5 of algorithm 1, where the regularization parameter can be tuned in addition to the kernel width parameter. In Williams et al. (2015b), the regularization is done implicitly through the pseudoinverse operation. We postulate that by manipulating the threshold used to decide which singular values are zero in the pseudoinverse computation, the performance of the kernel DMD method can be improved.

# 8 Conclusion

This manuscript revisits theoretical assumptions concerning DMD of Koopman operators, including the existence of lattices of eigenfunctions, common eigenfunctions between Koopman operators, and boundedness and compactness of Koopman operators. Counterexamples that illustrate restrictiveness of the assumptions are provided for each of the assumptions. In particular, a major theoretical result is established to show that the native RKHS of the Gaussian RBF kernel function only supports bounded Koopman operators if the dynamics are affine. Moreover, a kernel-based DMD algorithm that simplifies the algorithm in (Williams et al., 2015a) and presents it in an operator theoretic context is developed and validated through simulations.

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
