# OpenReview forum: "The Kernel Perspective on Dynamic Mode Decomposition"
_TMLR — Rejected by TMLR_

### Review · Reviewer_sZBJ · 2023-11-08

**Summary Of Contributions:**

The manuscript considers the theoretical properties of Koopman operators and suggests a novel kernel function that is better theoretically motivated. The counterexamples on the boundedness and compactness of Koopman operators, the existence of common eigenfunctions, and composing lattice from them are presented. Also, the authors state that the RBF kernel only induces the bounded Koopman operator for the affine dynamics. The presented DMD with a new kernel is tested in a classical benchmark cylinder flow dataset and visually shows the same result as the RBF kernel.

**Audience:**

Yes

**Broader Impact Concerns:**

No concerns on the ethical implications of the work.

**Claims And Evidence:**

No

**Requested Changes:**

Critical changes are listed below
1) The manuscript structure requires significant revision since the beginning of the text is just a review of advanced concepts from functional analysis and ergodic theory placed there without any explicitly stated motivation. The further text is also hard to read and extract the main contribution of the authors. I suggest highlighting the main intuition behind the abstract math concepts and using them to help the reader catch the addressed issues, the proposed solution, and the practical gain if any.
2) Moreover, although the authors provide artificial counterexamples for the Koopman operator properties, they do not illustrate what practical implications can be observed due to the violation of these properties. Thus, the practical motivation of the manuscript is unclear, please add the explicit problem statement section, where the quality metric is introduced and then used in the experiment section to support the gain from the proposed approach.
3) The single numerical test of the proposed new DMD framework is not enough for a solid justification of its practical importance. In addition, the presented test is not equipped with any quantities to estimate the approximation accuracy of the prediction generated by the baseline and the proposed kernel DMD. I do not observe any visual difference between generated snapshots. Therefore, the practical importance of the introduced kernel is not clear.

Changes to strengthen the work:
1) Check the typesetting: many typos, and wrong formatting. Unfortunately, without the line numbers listing typographic issues is impossible.
2) If the authors provide the modification of the DMD algorithm, it would be helpful to provide the pseudocode on how to transform snapshots to generate dynamic modes and how to use them further in forecasting
3) The link to the author's implementation of the proposed approach will be also very helpful for the community and simplify extensive testing of the proposed DMD approach in other, more complicated use cases

**Strengths And Weaknesses:**

**Strengths**
1) The manuscript has a solid theoretical basis and shows the formal proof of facts about the Koopman operator.
2) The counterexamples of Koopman operator properties are presented

**Weaknesses**
1) The text is hard to read. The motivation of this study and its practical implications are missed
2) The single numerical test is presented
3) This test does not show any (visual) difference between the proposed kernel and RBF
4) Experiments do not report any quantities to formally compare the considered DMD-like approaches
5) No ablation study in the experiments: how the proposed method is robust to the hyperparameter $\mu$ ?
6) The computational complexity (both asymptotic and runtime) of the introduced modification of Kernel DMD is completely ignored

---

### Review · Reviewer_RXz3 · 2023-12-10

**Summary Of Contributions:**

The authors present the theory of DMD of Koopman operators and their relationship to RKHS. The article also proposes a new DMD method with some simulations

**Audience:**

Yes

**Claims And Evidence:**

No

**Requested Changes:**

See my review above. In its current state, it is difficult to say what in particular needs to be changed. This papers needs to, at least:

- be specific about its contribution
- be clear about the proposed methodology, when it applies and how it differs from others
- provide convincing sufficient experimental validation

**Strengths And Weaknesses:**

I want to start with a disclaimer. I have worked with kernel methods and RKHS in the past, but in the last 5-7 years I have been working on other lines of research, so I am unfamiliar with the SOTA on the topic. Also, I am not an expert on DMD/Koopman, though I understand it.

The paper does a great job introducing the reader to the abovementioned objects; the unfamiliar reader would appreciate this attention.

However, I have the following reservations regarding the contribution of the paper:

-  It is difficult to assess where in the paper the contribution starts. The manuscript presents background material until (if I am not mistaken) page 7, and then the novel part starts in Sec 4. I am unsure if I captured the contribution to the extent that I cannot state that the contribution is at the bar of TMLR

- Sec 4 is said to "propose a method". However, it is not clear what this method is, how it differs from others, and its motivation. This is supposed to be the core contribution of the article; however, it is somewhat drowned out by the rest of the paper.

- The experimental validation is far from convincing. The authors only show a single synthetic example and then claim that **purely from visual inspection**, this example validates their proposal. This is a jumping conclusion not supported by the appropriate evidence.

- Not only does the paper lack a quantitative assessment of the method, but the experiment does not consider benchmarks or other methods for comparison.

- A particularly serious issue with this paper is that the Conclusion is almost a verbatim copy of the Abstract. Please avoid this practice

---

### Review · Reviewer_jovM · 2023-12-19

**Summary Of Contributions:**

The paper proposes a theoretical contribution related to the study of Koopman operators over RKHSs. It proves that if the Koopman operator of a real entire vector-valued function $F$ is bounded over the Gaussian RBF space, then $F$ is affine.

The paper also proposes an extension to the DMD algorithm for densely defined operators.

**Audience:**

Yes

**Claims And Evidence:**

No

**Requested Changes:**

- Fix notations. [critical]
- Improve the presentation of the derivations. [critical]
- Clarify the relaton with prior work [critical]
- Improve the part about vector-valued RKHSs.
- Provide a precise reference within Boas.

**Strengths And Weaknesses:**

The paper is of interest for the TMLR community. The theoretical contribution is sound and readers might be interested in the characterization of bounded Koopman operators over Gaussian RKHSs.

The numerical experiments complement nicely the theoretical derivations.

On the weak side, I found the paper hard to read for the following reasons:

It often uses semi-standard terminology that was not defined beforehand, examples of which include:
- order of a function
- type of a function
- \mathcal{I} for the imaginary part of a complex number
- the phrasing "Koopman generator"
- $\xi_{j, M}$ is never really defined.

Notation is also sometimes confusing to me, beyond the typos:
- The index of the observed states sometimes starts from $0$, sometimes from $1$, it would be nice to have some consistency throughout the paper.
- The formula (3) is not consistent with the text that precedes. I suggest to keep the use of subscripts $i$ and $j$ separated: $1 \leq i \leq m-1$ would be associated to the snapshots and $1 \leq j \leq M$ to the number of eigenvalues.
- The full state observable is denoted $g_{id}$ but does not depend on $d$, nor on $i$.
- The notation $(x)_d$ for the $d$-th coefficient of the vector $x$ is very heavy.
- The vector-valued eigenfunction is denoted $\phi_{s, i}$ but it does not depend on $i$ either.

The paper is structured using a lot of inline mathematics, which makes it hard to follow the logic. The reader (at least me) never really knows what is part of previous published work, and what corresponds to the derivations of the authors. The statements are imprecise (because they are mostly in text blocks, not in lemmas) and the assumptions are inconsistent, we never know under which setting we must understand the derivations that are displayed. In my opinion, the paper is lacking a clear set of assumptions which together with a better structure would make crystal clear the setting at a given line. Example:
- For the proposed DMD, $x \mapsto x_d$ is assumed to be in the RKHS, but that rarely happens. What happens with the gaussian kernel which violates this assumption ?
- The critical passage that explicits $[P_\alpha \mathcal{K}_F]_{\alpha}^{\alpha}$ deserves more explanation, a lemma would be welcome.

The part about vv-RKHS could benefit from more developments, and I struggle to see a real contribution in this section.
- From what I understand, the proposed vv-RKHS approach is just a rewriting of the previous approach that consists in estimating separately the different components. The text is a bit misleading, as it states that until this work methods have simply learned the individual components separately: this is true, but this paper also takes this approach due to the choice of the kernel. This is due to the operator-valued kernel (ovk) being equal to $k_{\text{scalar}} \cdot I_n$ where $I_n$ is the identity matrix of size $n \times n$, which in turn gives a specific structure to the vv-RKHS that is a product of real RKHSs with inner product obtained naturally by adding the inner products of each components.
- References to the operator-valued kernel literature is missing, I suggest the classical [1].
- It would be interesting to extend this work to decomposable ovks with matrix parts different from identity. $G$ would then lose its block-diagonal structure but would have a tensorial structure that can be exploited for the task.
- When working with both scalar and operator-valued kernel, it can be worth it to use lower cases for scalar quantities and upper cases for vector-valued ones, as it makes the reader visually aware of what is what in a mathematical expression.

[1] Carmeli, Claudio, et al. "Vector valued reproducing kernel Hilbert spaces and universality." Analysis and Applications 8.01 (2010): 19-61.

Questions/Remarks:

- What is the point of the discussion about universal kernels ? It is never mentioned past the first section, and there is no consistency theorem, nor even a discussion that needs to have this notion introduced.
- What is the argument for saying that the $\xi_{j,M}$ exist, without assumptions on the family of $(\phi_j)_{j=1}^\infty$ ?
- Could you explicit the difference between [Williams 2015a] and your approach for DMD ?
- In section 2, the fact that $\hat{\phi}$ is a uniform $\epsilon$-approximation of an eigenvector cannot imply the given equality $\hat{\phi}(x_{i+1}) = \lambda^{i+1} \hat{\phi}(x_0) + \frac{1 - \lambda^{i+1}}{1 - \lambda} \cdot \epsilon$ but should be rephrased as an inequality.

Typos:
- Lemma 3: "over a the" -> remove the "a"
- In the text before equation (3), when writing the decomposition of $(x)_d$, there is a typo on the index: $\phi_i$ should be $\phi_j$.

---

### Decision · Action_Editor_vpdn · 2024-02-25

**Recommendation:** Reject

**Comment:**

1) All reviewers agree that paper is not easy to read and follow, and motivation behind the study is not well-presented.
2)The paper has some merits, but the proposed new algorithms are weak from the numerical side, for example, it is tested only on one synthetic examples and visual comparison is presented.

**Audience:**

Koopman operator receives reasonable attention among people working with ML for dynamical systems, probably the main audience.

**Claims And Evidence:**

The paper provides a theoretical study of Koopman operators over Reproducing Kernel Hilbert spaces: if Koopman operator for F is bounded, then F is affine. They also propose a new DMD method.